# Without-Replacement Sampling
# for Stochastic Gradient Methods

**Ohad Shamir**
Department of Computer Science and Applied Mathematics
Weizmann Institute of Science
Rehovot, Israel
ohad.shamir@weizmann.ac.il

## Abstract

Stochastic gradient methods for machine learning and optimization problems are usually analyzed assuming data points are sampled *with* replacement. In contrast, sampling *without* replacement is far less understood, yet in practice it is very common, often easier to implement, and usually performs better. In this paper, we provide competitive convergence guarantees for without-replacement sampling under several scenarios, focusing on the natural regime of few passes over the data. Moreover, we describe a useful application of these results in the context of distributed optimization with randomly-partitioned data, yielding a nearly-optimal algorithm for regularized least squares (in terms of both communication complexity and runtime complexity) under broad parameter regimes. Our proof techniques combine ideas from stochastic optimization, adversarial online learning and transductive learning theory, and can potentially be applied to other stochastic optimization and learning problems.

## 1 Introduction

Many canonical machine learning problems boil down to solving a convex empirical risk minimization problem of the form

$$\min_{\mathbf{w} \in \mathcal{W}} F(\mathbf{w}) = \frac{1}{m} \sum_{i=1}^{m} f_i(\mathbf{w}), \tag{1}$$

where each individual function $f_i(\cdot)$ is convex (e.g. the loss on a given example in the training data), and the set $\mathcal{W} \subseteq \mathbb{R}^d$ is convex. In large-scale applications, where both $m, d$ can be huge, a very popular approach is to employ stochastic gradient methods. Generally speaking, these methods maintain some iterate $\mathbf{w}_t \in \mathcal{W}$, and at each iteration, sample an individual function $f_i(\cdot)$, and perform some update to $\mathbf{w}_t$ based on $\nabla f_i(\mathbf{w}_t)$. Since the update is with respect to a single function, this update is usually computationally cheap. Moreover, when the sampling is done independently and uniformly at random, $\nabla f_i(\mathbf{w}_t)$ is an unbiased estimator of the true gradient $\nabla F(\mathbf{w}_t)$, which allows for good convergence guarantees after a reasonable number of iterations (see for instance [18, 15]).

However, in practical implementations of such algorithms, it is actually quite common to use *without-replacement* sampling, or equivalently, pass sequentially over a random shuffling of the functions $f_i$. Intuitively, this forces the algorithm to process more equally all data points, and often leads to better empirical performance. Moreover, without-replacement sampling is often easier and faster to implement, as it requires sequential data access, as opposed to the random data access required by with-replacement sampling (see for instance [3, 16, 8]).

## 1.1 What is Known so Far?

Unfortunately, without-replacement sampling is not covered well by current theory. The challenge is that unlike with-replacement sampling, the functions processed at every iteration are not statistically independent, and their correlations are difficult to analyze. Since this lack of theory is the main motivation for our paper, we describe the existing known results in some detail, before moving to our contributions.

To begin with, there exist classic convergence results which hold deterministically for every order in which the individual functions are processed, and in particular when we process them by sampling without replacement (e.g. [14]). However, these can be exponentially worse than those obtained using random without-replacement sampling, and this gap is inevitable (see for instance [16]).

More recently, Recht and Ré [16] studied this problem, attempting to show that at least for least squares optimization, without-replacement sampling is always better (or at least not substantially worse) than with-replacement sampling on a given dataset. They showed this reduces to a fundamental conjecture about arithmetic-mean inequalities for matrices, and provided partial results in that direction, such as when the individual functions themselves are assumed to be generated i.i.d. from some distribution. However, the general question remains open.

In a recent breakthrough, Gürbüzbalaban et al. [8] provided a new analysis of gradient descent algorithms for solving Eq. (1) based on random reshuffling: Each epoch, the algorithm draws a new permutation on $\{1, \ldots, m\}$ uniformly at random, and processes the individual functions in that order. Under smoothness and strong convexity assumptions, the authors obtain convergence guarantees of essentially $\mathcal{O}(1/k^2)$ after $k$ epochs, vs. $\mathcal{O}(1/k)$ using with-replacement sampling (with the $\mathcal{O}(\cdot)$ notation including certain dependencies on the problem parameters and data size). Thus, without-replacement sampling is shown to be strictly better than with-replacement sampling, after sufficiently many passes over the data. However, this leaves open the question of why without-replacement sampling works well after a few – or even just one – passes over the data. Indeed, this is often the regime at which stochastic gradient methods are most useful, do not require repeated data reshuffling, and their good convergence properties are well-understood in the with-replacement case.

## 1.2 Our Results

In this paper, we provide convergence guarantees for stochastic gradient methods, under several scenarios, in the natural regime where the number of passes over the data is small, and in particular that no data reshuffling is necessary. We emphasize that our goal here will be more modest than those of [16, 8]: Rather than show superiority of without-replacement sampling, we only show that it will not be significantly worse (in a worst-case sense) than with-replacement sampling. Nevertheless, such guarantees are novel, and still justify the use of with-replacement sampling, especially in situations where it is advantageous due to data access constraints or other reasons. Moreover, these results have a useful application in the context of distributed learning and optimization, as we will shortly describe. Our main contributions can be summarized as follows:

- For convex functions on some convex domain $\mathcal{W}$, we consider algorithms which perform a single pass over a random permutation of $m$ individual functions, and show that their suboptimality can be characterized by a combination of two quantities, each from a different field: First, the *regret* which the algorithm can attain in the setting of *adversarial online convex optimization* [17, 9], and second, the *transductive Rademacher complexity* of $\mathcal{W}$ with respect to the individual functions, a notion stemming from transductive learning theory [22, 6].

- As a concrete application of the above, we show that if each function $f_i(\cdot)$ corresponds to a convex Lipschitz loss of a linear predictor, and the algorithm belongs to the class of algorithms which in the online setting attain $\mathcal{O}(\sqrt{T})$ regret on $T$ such functions (which includes, for example, stochastic gradient descent), then the suboptimality using without-replacement sampling, after processing $T$ functions, is $\mathcal{O}(1/\sqrt{T})$. Up to numerical constants, the guarantee is the same as that obtained using with-replacement sampling.

- We turn to consider more specifically the stochastic gradient descent algorithm, and show that if the objective function $F(\cdot)$ is $\lambda$-strongly convex, and the functions $f_i(\cdot)$ are also smooth, then the suboptimality bound becomes $\mathcal{O}(1/\lambda T)$, which matches the with-replacement guarantees

(although with replacement, smoothness is not needed, and the dependence on some parameters hidden in the $\mathcal{O}(\cdot)$ is somewhat better).

- In recent years, a new set of fast stochastic algorithms to solve Eq. (1) has emerged, such as SAG, SDCA, SVRG, and quite a few other variants. These algorithms are characterized by cheap stochastic iterations, involving computations of individual function gradients, yet unlike traditional stochastic algorithms, enjoy a linear convergence rate (runtime scaling logarithmically with the required accuracy). To the best of our knowledge, all existing analyses require sampling with replacement. We consider a representative algorithm from this set, namely SVRG, and the problem of regularized least squares, and show that similar guarantees can be obtained using without-replacement sampling. This result has a potentially interesting implication: Under the mild assumption that the problem's condition number is smaller than the data size, we get that SVRG can converge to an arbitrarily accurate solution (even up to machine precision), *without* the need to reshuffle the data – only a single shuffle at the beginning suffices. Thus, at least for this problem, we can obatin fast and high-quality solutions even if random data access is expensive.

- A further application of the SVRG result is in the context of distributed learning: By simulating without-replacement SVRG on data randomly partitioned between several machines, we get a nearly-optimal algorithm for regularized least squares, in terms of communication and computational complexity, as long as the condition number is smaller than the data size per machine (up to logarithmic factors). This builds on the work of Lee et al. [13], who were the first to recognize the applicability of SVRG to distributed optimization. However, their results relied on with-replacement sampling, and are applicable only for much smaller condition numbers.

We note that our focus is on scenarios where no reshufflings are necessary. In particular, the $\mathcal{O}(1/\sqrt{T})$ and $\mathcal{O}(1/\lambda T)$ bounds apply for all $T \in \{1, 2, \ldots, m\}$, namely up to one full pass over a random permutation of the entire data. However, our techniques are also applicable to a constant $(> 1)$ number of passes, by randomly reshuffling the data after every pass. In a similar vein, our SVRG result can be readily extended to a situation where each epoch of the algorithm is done on an independent permutation of the data. We leave a full treatment of this to future work.

## 2 Preliminaries and Notation

We use bold-face symbols to denote vectors. Given a vector $\mathbf{w}$, $w_i$ denotes it's $i$-th coordinate. We utilize the standard $\mathcal{O}(\cdot), \Theta(\cdot), \Omega(\cdot)$ notation to hide constants, and $\tilde{\mathcal{O}}, \tilde{\Theta}(\cdot), \tilde{\Omega}(\cdot)$ to hide constants and logarithmic factors.

Given convex functions $f_1(\cdot), f_2(\cdot), \ldots, f_m(\cdot)$ from $\mathbb{R}^d$ to $\mathbb{R}$, we define our objective function $F : \mathbb{R}^d \to \mathbb{R}$ as

$$F(\mathbf{w}) = \frac{1}{m} \sum_{i=1}^{m} f_i(\mathbf{w}),$$

with some fixed optimum $\mathbf{w}^* \in \arg\min_{\mathbf{w} \in \mathcal{W}} F(\mathbf{w})$. In machine learning applications, each individual $f_i(\cdot)$ usually corresponds to a loss with respect to a data point, hence will use the terms "individual function", "loss function" and "data point" interchangeably throughout the paper.

We let $\sigma$ be a permutation over $\{1, \ldots, m\}$ chosen uniformly at random. In much of the paper, we consider methods which draw loss functions without replacement according to that permutation (that is, $f_{\sigma(1)}(\cdot), f_{\sigma(2)}(\cdot), f_{\sigma(3)}(\cdot), \ldots$). We will use the shorthand notation

$$F_{1:t-1}(\cdot) = \frac{1}{t-1} \sum_{i=1}^{t-1} f_{\sigma(i)}(\cdot) \;\; , \;\; F_{t:m}(\cdot) = \frac{1}{m-t+1} \sum_{i=t}^{m} f_{\sigma(i)}(\cdot)$$

to denote the average loss with respect to the first $t - 1$ and last $m - t + 1$ loss functions respectively, as ordered by the permutation (intuitively, the losses in $F_{1:t-1}(\cdot)$ are those already observed by the algorithm at the beginning of iteration $t$, whereas the losses in $F_{t:m}(\cdot)$ are those not yet observed). We use the convention that $F_{1:1}(\cdot) \equiv 0$, and the same goes for other expressions of the form $\frac{1}{t-1} \sum_{i=1}^{t-1} \cdots$ throughout the paper, when $t = 1$. We also define quantities such as $\nabla F_{1:t-1}(\cdot)$ and $\nabla F_{t:m}(\cdot)$ similarly.

A function $f : \mathbb{R}^d \rightarrow \mathbb{R}$ is $\lambda$-*strongly convex*, if for any $\mathbf{w}, \mathbf{w}'$, $f(\mathbf{w}') \geq f(\mathbf{w}) + \langle \mathbf{g}, \mathbf{w}' - \mathbf{w} \rangle + \frac{\lambda}{2} \|\mathbf{w}' - \mathbf{w}\|^2$, where $\mathbf{g}$ is any (sub)-gradient of $f$ at $\mathbf{w}$, and is $\mu$-*smooth* if for any $\mathbf{w}, \mathbf{w}'$, $f(\mathbf{w}') \leq f(\mathbf{w}) + \langle \mathbf{g}, \mathbf{w}' - \mathbf{w} \rangle + \frac{\mu}{2} \|\mathbf{w}' - \mathbf{w}\|^2$. $\mu$-smoothness also implies that the function $f$ is differentiable, and its gradient is $\mu$-Lipschitz. Based on these properties, it is easy to verify that if $\mathbf{w}^* \in \arg\min f(\mathbf{w})$, and $f$ is $\lambda$-strongly convex and $\mu$-smooth, then $\frac{\lambda}{2} \|\mathbf{w} - \mathbf{w}^*\|^2 \leq f(\mathbf{w}) - f(\mathbf{w}^*) \leq \frac{\mu}{2} \|\mathbf{w} - \mathbf{w}^*\|^2$.

We will also require the notion of trandsuctive Rademacher complexity, as developed by El-Yaniv and Pechyony [6, Definition 1], with a slightly different notation adapted to our setting:

**Definition 1.** *Let $\mathcal{V}$ be a set of vectors $\mathbf{v} = (v_1, \ldots, v_m)$ in $\mathbb{R}^m$. Let $s, u$ be positive integers such that $s + u = m$, and denote $p := \frac{su}{(s+u)^2} \in (0, 1/2)$. We define the transductive Rademacher Complexity $\mathcal{R}_{s,u}(\mathcal{V})$ as $\mathcal{R}_{s,u}(\mathcal{V}) = \left(\frac{1}{s} + \frac{1}{u}\right) \cdot \mathbb{E}_{r_1,\ldots,r_m} \left[ \sup_{\mathbf{v} \in \mathcal{V}} \sum_{i=1}^{m} r_i v_i \right]$, where $r_1, \ldots, r_m$ are i.i.d. random variables such that $r_i = 1$ or $-1$ with probability $p$, and $r_i = 0$ with probability $1 - 2p$.*

This quantity is an important parameter is studying the richness of the set $\mathcal{V}$, and will prove crucial in providing some of the convergence results presented later on. Note that it differs slightly from standard Rademacher complexity, which is used in the theory of learning from i.i.d. data, where the Rademacher variables $r_i$ only take $-1, +1$ values, and $(1/s + 1/u)$ is replaced by $1/m$).

## 3 Convex Lipschitz Functions

We begin by considering loss functions $f_1(\cdot), f_2(\cdot), \ldots, f_m(\cdot)$ which are convex and $L$-Lipschitz over some convex domain $\mathcal{W}$. We assume the algorithm sequentially goes over some permuted ordering of the losses, and before processing the $t$-th loss function, produces an iterate $\mathbf{w}_t \in \mathcal{W}$. Moreover, we assume that the algorithm has a regret bound in the adversarial online setting, namely that for *any* sequence of $T$ convex Lipschitz losses $f_1(\cdot), \ldots, f_T(\cdot)$, and any $\mathbf{w} \in \mathcal{W}$,

$$\sum_{t=1}^{T} f_t(\mathbf{w}_t) - \sum_{t=1}^{T} f_t(\mathbf{w}) \leq R_T$$

for some $R_T$ scaling sub-linearly in $T$[1]. For example, online gradient descent (which is equivalent to stochastic gradient descent when the losses are i.i.d.), with a suitable step size, satisfies $R_T = \mathcal{O}(BL\sqrt{T})$, where $L$ is the Lipschitz parameter and $B$ upper bounds the norm of any vector in $\mathcal{W}$. A similar regret bound can also be shown for other online algorithms (see [9, 17, 23]).

Since the ideas used for analyzing this setting will also be used in the more complicated results which follow, we provide the analysis in some detail. We first have the following straightforward theorem, which upper bounds the expected suboptimality in terms of regret and the expected difference between the average loss on prefixes and suffixes of the data.

**Theorem 1.** *Suppose the algorithm has a regret bound $R_T$, and sequentially processes $f_{\sigma(1)}(\cdot), \ldots, f_{\sigma(T)}(\cdot)$ where $\sigma$ is a random permutation on $\{1, \ldots, m\}$. Then in expectation over $\sigma$,*

$$\mathbb{E}\left[ \frac{1}{T} \sum_{t=1}^{T} F(\mathbf{w}_t) - F(\mathbf{w}^*) \right] \leq \frac{R_T}{T} + \frac{1}{mT} \sum_{t=2}^{T} (t-1) \cdot \mathbb{E}[F_{1:t-1}(\mathbf{w}_t) - F_{t:m}(\mathbf{w}_t)].$$

The left hand side in the inequality above can be interpreted as an expected bound on $F(\mathbf{w}_t) - F(\mathbf{w}^*)$, where $t$ is drawn uniformly at random from $1, 2, \ldots, T$. Alternatively, by Jensen's inequality and the fact that $F(\cdot)$ is convex, the same bound also applies on $\mathbb{E}[F(\bar{\mathbf{w}}_T) - F(\mathbf{w}^*)]$, where $\bar{\mathbf{w}}_T = \frac{1}{T} \sum_{t=1}^{T} \mathbf{w}_t$.

The proof of the theorem relies on the following simple but key lemma, which expresses the expected difference between with-replacement and without-replacement sampling in an alternative form, similar to Thm. 1 and one which lends itself to tools and ideas from transductive learning theory. This lemma will be used in proving all our main results, and its proof appears in Subsection A.2

**Lemma 1.** *Let $\sigma$ be a permutation over $\{1, \ldots, m\}$ chosen uniformly at random. Let $s_1, \ldots, s_m \in \mathbb{R}$ be random variables which conditioned on $\sigma(1), \ldots, \sigma(t-1)$, are independent of $\sigma(t), \ldots, \sigma(m)$. Then $\mathbb{E}\left[\frac{1}{m}\sum_{i=1}^m s_i - s_{\sigma(t)}\right]$ equals $\frac{t-1}{m} \cdot \mathbb{E}\left[s_{1:t-1} - s_{t:m}\right]$ for $t > 1$, and $0$ for $t = 1$.*

*Proof of Thm. 1.* Adding and subtracting terms, and using the facts that $\sigma$ is a permutation chosen uniformly at random, and $\mathbf{w}^*$ is fixed,

$$\mathbb{E}\left[\frac{1}{T}\sum_{t=1}^T F(\mathbf{w}_t) - F(\mathbf{w}^*)\right] = \mathbb{E}\left[\frac{1}{T}\sum_{t=1}^T \left(f_{\sigma(t)}(\mathbf{w}_t) - F(\mathbf{w}^*)\right)\right] + \mathbb{E}\left[\frac{1}{T}\sum_{t=1}^T \left(F(\mathbf{w}_t) - f_{\sigma(t)}(\mathbf{w}_t)\right)\right]$$

$$= \mathbb{E}\left[\frac{1}{T}\sum_{t=1}^T \left(f_{\sigma(t)}(\mathbf{w}_t) - f_{\sigma(t)}(\mathbf{w}^*)\right)\right] + \mathbb{E}\left[\frac{1}{T}\sum_{t=1}^T \left(F(\mathbf{w}_t) - f_{\sigma(t)}(\mathbf{w}_t)\right)\right]$$

Applying the regret bound assumption on the sequence of losses $f_{\sigma(1)}(\cdot), \ldots, f_{\sigma(T)}(\cdot)$, the above is at most $\frac{R_T}{T} + \frac{1}{T}\sum_{t=1}^T \mathbb{E}\left[F(\mathbf{w}_t) - f_{\sigma(t)}(\mathbf{w}_t)\right]$. Since $\mathbf{w}_t$ (as a random variable over the permutation $\sigma$ of the data) depends only on $\sigma(1), \ldots, \sigma(t-1)$, we can use Lemma 1 (where $s_i = f_i(\mathbf{w}_t)$, and noting that the expectation above is 0 when $t = 1$), and get that the above equals $\frac{R_T}{T} + \frac{1}{mT}\sum_{t=2}^T (t-1) \cdot \mathbb{E}[F_{1:t-1}(\mathbf{w}_t) - F_{t:m}(\mathbf{w}_t)]$. $\square$

Having reduced the expected suboptimality to the expected difference $\mathbb{E}[F_{1:t-1}(\mathbf{w}_t) - F_{t:m}(\mathbf{w}_t)]$, the next step is to upper bound it with $\mathbb{E}[\sup_{\mathbf{w}\in\mathcal{W}}(F_{1:t-1}(\mathbf{w}) - F_{t:m}(\mathbf{w}))]$: Namely, having split our loss functions at random to two groups of size $t-1$ and $m-t+1$, how large can be the difference between the average loss of any $\mathbf{w}$ on the two groups? Such uniform convergence bounds are exactly the type studied in transductive learning theory, where a fixed dataset is randomly split to a training set and a test set, and we consider the generalization performance of learning algorithms ran on the training set. Such results can be provided in terms of the transductive Rademacher complexity of $\mathcal{W}$, and combined with Thm. 1, lead to the following bound in our setting:

**Theorem 2.** *Suppose that each $\mathbf{w}_t$ is chosen from a fixed domain $\mathcal{W}$, that the algorithm enjoys a regret bound $R_T$, and that $\sup_{i,\mathbf{w}\in\mathcal{W}} |f_i(\mathbf{w})| \leq B$. Then in expectation over the random permutation $\sigma$,*

$$\mathbb{E}\left[\frac{1}{T}\sum_{t=1}^T F(\mathbf{w}_t) - F(\mathbf{w}^*)\right] \leq \frac{R_T}{T} + \frac{1}{mT}\sum_{t=2}^T (t-1)\mathcal{R}_{t-1:m-t+1}(\mathcal{V}) + \frac{24B}{\sqrt{m}},$$

*where $\mathcal{V} = \{(f_1(\mathbf{w}), \ldots, f_m(\mathbf{w})) \mid \mathbf{w} \in \mathcal{W}\}$.*

Thus, we obtained a generic bound which depends on the online learning characteristics of the algorithm, as well as the statistical learning properties of the class $\mathcal{W}$ on the loss functions. The proof (as the proofs of all our results from now on) appears in Section A.

We now instantiate the theorem to the prototypical case of bounded-norm linear predictors, where the loss is some convex and Lipschitz function of the prediction $\langle \mathbf{w}, \mathbf{x} \rangle$ of a predictor $\mathbf{w}$ on an instance $\mathbf{x}$, possibly with some regularization:

**Corollary 1.** *Under the conditions of Thm. 2, suppose that $\mathcal{W} \subseteq \{\mathbf{w} : \|\mathbf{w}\| \leq \bar{B}\}$, and each loss function $f_i$ has the form $\ell_i(\langle \mathbf{w}, \mathbf{x}_i \rangle) + r(\mathbf{w})$ for some $L$-Lipschitz $\ell_i$, $\|\mathbf{x}_i\| \leq 1$, and a fixed function $r$. Then $\mathbb{E}\left[\frac{1}{T}\sum_{t=1}^T F(\mathbf{w}_t) - F(\mathbf{w}^*)\right] \leq \frac{R_T}{T} + \frac{2(12+\sqrt{2})\bar{B}L}{\sqrt{m}}$.*

As discussed earlier, in the setting of Corollary 1, typical regret bounds are on the order of $\mathcal{O}(\bar{B}L\sqrt{T})$. Thus, the expected suboptimality is $\mathcal{O}(\bar{B}L/\sqrt{T})$, all the way up to $T = m$ (i.e. a full pass over a random permutation of the data). Up to constants, this is the same as the suboptimality attained by $T$ iterations of with-replacement sampling, using stochastic gradient descent or similar algorithms.

## 4 Faster Convergence for Strongly Convex Functions

We now consider more specifically the stochastic gradient descent algorithm on a convex domain $\mathcal{W}$, which can be described as follows: We initialize at some $\mathbf{w}_1 \in \mathcal{W}$, and perform the update steps

$$\mathbf{w}_{t+1} = \Pi_\mathcal{W}(\mathbf{w}_t - \eta_t \mathbf{g}_t),$$

where $\eta_t > 0$ are fixed step sizes, $\Pi_\mathcal{W}$ is projection on $\mathcal{W}$, and $\mathbf{g}_t$ is a subgradient of $f_{\sigma(t)}(\cdot)$ at $\mathbf{w}_t$. Moreover, we assume the objective function $F(\cdot)$ is $\lambda$-strongly convex for some $\lambda > 0$. In this scenario, using with-replacement sampling (i.e. $\mathbf{g}_t$ is a subgradient of an independently drawn $f_i(\cdot)$), performing $T$ iterations as above and returning a randomly sampled iterate $\mathbf{w}_t$ or their average results in expected suboptimality $\tilde{\mathcal{O}}(G^2/\lambda T)$, where $G^2$ is an upper bound on the expected squared norm of $\mathbf{g}_t$ [15, 18]. Here, we study a similar situation in the without-replacement case.

In the result below, we will consider specifically the case where each $f_i(\mathbf{w})$ is a Lipschitz and smooth loss of a linear predictor $\mathbf{w}$, possibly with some regularization. The smoothness assumption is needed to get a good bound on the transductive Rademacher complexity of quantities such as $\langle \nabla f_i(\mathbf{w}), \mathbf{w} \rangle$. However, the technique can be potentially applied to more general cases.

**Theorem 3.** *Suppose $\mathcal{W}$ has diameter $B$, and that $F(\cdot)$ is $\lambda$-strongly convex on $\mathcal{W}$. Assume that $f_i(\mathbf{w}) = \ell_i(\langle \mathbf{w}, \mathbf{x}_i \rangle) + r(\mathbf{w})$ where $\|\mathbf{x}_i\| \leq 1$, $r(\cdot)$ is possibly some regularization term, and each $\ell_i$ is L-Lipschitz and $\mu$-smooth on $\{z : z = \langle \mathbf{w}, \mathbf{x} \rangle, \mathbf{w} \in \mathcal{W}, \|\mathbf{x}\| \leq 1\}$. Furthermore, suppose $\sup_{\mathbf{w} \in \mathcal{W}} \|\nabla f_i(\mathbf{w})\| \leq G$. Then for any $1 < T \leq m$, if we run SGD for $T$ iterations with step size $\eta_t = 2/\lambda t$, we have (for a universal positive constant $c$)*

$$\mathbb{E}\left[\frac{1}{T}\sum_{t=1}^{T} F(\mathbf{w}_t) - F(\mathbf{w}^*)\right] \leq c \cdot \frac{((L + \mu B)^2 + G^2)\log(T)}{\lambda T}.$$

As in the results of the previous section, the left hand side is the expected optimality of a single $\mathbf{w}_t$ where $t$ is chosen uniformly at random, or an upper bound on the expected suboptimality of the average $\bar{\mathbf{w}}_T = \frac{1}{T}\sum_{t=1}^{T} \mathbf{w}_t$. This result is similar to the with-replacement case, up to numerical constants and the additional $(L + \mu B^2)$ term in the numerator. We note that in some cases, $G^2$ is the dominant term anyway[2]. However, it is not clear that the $(L + \mu B^2)$ term is necessary, and removing it is left to future work. We also note that the $\log(T)$ factor in the theorem can be removed by considering not $\frac{1}{T}\sum_{t=1}^{T} F(\mathbf{w}_t)$, but rather only an average over some suffix of the iterates, or weighted averaging (see for instance [15, 12, 21], where the same techniques can be applied here).

The proof of Thm. 3 is somewhat more challenging than the results of the previous section, since we are attempting to get a faster rate of $\mathcal{O}(1/T)$ rather than $\mathcal{O}(1/\sqrt{T})$, all the way up to $T = m$. A significant technical obstacle is that our proof technique relies on concentration of averages around expectations, which on $T$ samples does not go down faster than $\mathcal{O}(1/\sqrt{T})$. To overcome this, we apply concentration results not on the function values (i.e. $F_{1:t-1}(\mathbf{w}_t) - F_{t:m}(\mathbf{w}_t)$ as in the previous section), but rather on gradient-iterate inner products, i.e. $\langle \nabla F_{1:t-1}(\mathbf{w}_t) - \nabla F_{t:m}(\mathbf{w}_t), \mathbf{w}_t - \mathbf{w}^* \rangle$, where $\mathbf{w}^*$ is the optimal solution. To get good bounds, we need to assume these gradients have a certain structure, which is why we need to make the assumption in the theorem about the form of each $f_i(\cdot)$. Using transductive Rademacher complexity tools, we manage to upper bound the expectation of these inner products by quantities roughly of the form $\sqrt{\mathbb{E}[\|\mathbf{w}_t - \mathbf{w}^*\|^2]}/\sqrt{t}$ (assuming here $t < m/2$ for simplicity). We now utilize the fact that in the strongly convex case, $\|\mathbf{w}_t - \mathbf{w}^*\|$ itself decreases to zero with $t$ at a certain rate, to get fast rates decreasing as $1/t$.

## 5  Without-Replacement SVRG for Least Squares

In this section, we will consider a more sophisticated stochastic gradient approach, namely the SVRG algorithm of [11], designed to solve optimization problems with a finite sum structure as in Eq. (1). Unlike purely stochastic gradient procedures, this algorithm does require several passes over the data. However, assuming the condition number $1/\lambda$ is smaller than the data size (assuming each $f_i(\cdot)$ is $\mathcal{O}(1)$ smooth, and $\lambda$ is the strong convexity parameter of $F(\cdot)$), only $\mathcal{O}(m\log(1/\epsilon))$ gradient evaluations are required to get an $\epsilon$-accurate solution, for any $\epsilon$. Thus, we can get a high-accuracy solution after the equivalent of a small number of data passes. As discussed in the introduction, over the past few years several other algorithms have been introduced and shown to have such a behavior. We will focus on the algorithm in its basic form, where the domain $\mathcal{W}$ equals $\mathbb{R}^d$.

The existing analysis of SVRG ([11]) assumes stochastic iterations, which involves sampling the data with replacement. Thus, it is natural to consider whether a similar convergence behavior occurs using

without-replacement sampling. As we shall see, a positive reply has at least two implications: The first is that as long as the condition number is smaller than the data size, SVRG can be used to obtain a high-accuracy solution, *without* the need to reshuffle the data: Only a single shuffle at the beginning suffices, and the algorithm terminates after a small number of *sequential* passes (logarithmic in the required accuracy). The second implication is that such without-replacement SVRG can be used to get a nearly-optimal algorithm for convex distributed learning and optimization on randomly partitioned data, as long as the condition number is smaller than the data size at each machine.

The SVRG algorithm using without-replacement sampling on a dataset of size $m$ is described as Algorithm 1. It is composed of multiple epochs (indexed by $s$), each involving one gradient computation on the entire dataset, and $T$ stochastic iterations, involving gradient computations with respect to individual data points. Although the gradient computation on the entire dataset is expensive, it is only needed to be done once per epoch. Since the algorithm will be shown to have linear convergence as a function of the number of epochs, this requires only a small (logarithmic) number of passes over the data.

---

**Algorithm 1** SVRG using Without-Replacement Sampling

> **Parameters:** $\eta, T$, permutation $\sigma$ on $\{1, \ldots, m\}$
> Initialize $\tilde{\mathbf{w}}_1$ at $\mathbf{0}$
> **for** $s = 1, 2, \ldots$ **do**
> $\quad \mathbf{w}_{(s-1)T+1} := \tilde{\mathbf{w}}_s$
> $\quad \tilde{\mathbf{n}} := \nabla F(\tilde{\mathbf{w}}_s) = \frac{1}{m} \sum_{i=1}^{m} \nabla f_i(\tilde{\mathbf{w}}_s)$
> $\quad$ **for** $t = (s-1)T + 1, \ldots, sT$ **do**
> $\quad\quad \mathbf{w}_{t+1} := \mathbf{w}_t - \eta \left( \nabla f_{\sigma(t)}(\mathbf{w}_t) - \nabla f_{\sigma(t)}(\tilde{\mathbf{w}}_s) + \tilde{\mathbf{n}} \right)$
> $\quad$ **end for**
> $\quad$ Let $\tilde{\mathbf{w}}_{s+1}$ be the average of $\mathbf{w}_{(s-1)T+1}, \ldots, \mathbf{w}_{sT}$, or one of them drawn uniformly at random.
> **end for**

---

We will consider specifically the regularized least mean squares problem, where

$$f_i(\mathbf{w}) = \frac{1}{2} \left( \langle \mathbf{w}, \mathbf{x}_i \rangle - y_i \right)^2 + \frac{\hat{\lambda}}{2} \left\| \mathbf{w} \right\|^2 . \tag{2}$$

for some $\mathbf{x}_i, y_i$ and $\hat{\lambda} > 0$. Moreover, we assume that $F(\mathbf{w}) = \frac{1}{m} \sum_{i=1}^{m} f_i(\mathbf{w})$ is $\lambda$-strongly convex (note that necessarily $\lambda \geq \hat{\lambda}$). For convenience, we will assume that $\|\mathbf{x}_i\|, |y_i|, \lambda$ are all at most 1 (this is without much loss of generality, since we can always re-scale the loss functions by an appropriate factor). Note that under this assumption, each $f_i(\cdot)$ as well as $F(\cdot)$ are also $1 + \hat{\lambda} \leq 2$-smooth.

**Theorem 4.** *Suppose each loss function $f_i(\cdot)$ corresponds to Eq. (2), where $\mathbf{x}_i \in \mathbb{R}^d$, $\max_i \|\mathbf{x}_i\| \leq 1$, $\max_i |y_i| \leq 1$, $\hat{\lambda} > 0$, and that $F(\cdot)$ is $\lambda$-strongly convex, where $\lambda \in (0, 1)$. Moreover, let $B \geq 1$ be such that $\|\mathbf{w}^*\|^2 \leq B$ and $\max_t F(\mathbf{w}_t) - F(\mathbf{w}^*) \leq B$ with probability 1 over the random permutation. There is some universal constant $c_0 \geq 1$, such that for any $c \geq c_0$ and any $\epsilon \in (0, 1)$, if we run algorithm 1, using parameters $\eta, T$ satisfying*

$$\eta = \frac{1}{c} \;\; , \;\; T \geq \frac{9}{\eta \lambda} \;\; , \;\; m \geq c \log^2 \left( \frac{64 dm B^2}{\lambda \epsilon} \right) T,$$

*then after $S = \lceil \log_4(9/\epsilon) \rceil$ epochs of the algorithm, $\tilde{\mathbf{w}}_{S+1}$ satisfies $\mathbb{E}[F(\tilde{\mathbf{w}}_{S+1}) - \min_{\mathbf{w}} F(\mathbf{w})] \leq \epsilon$.*

In particular, by taking $\eta = \Theta(1)$ and $T = \Theta(1/\lambda)$, the algorithm attains an $\epsilon$-accurate solution after $\mathcal{O}(\log(1/\epsilon)/\lambda)$ stochastic iterations of without-replacement sampling, and $\mathcal{O}(\log(1/\epsilon))$ sequential passes over the data to compute gradients of $F(\cdot)$. This implies that as long as $1/\lambda$ (which stands for the condition number of the problem) is smaller than $\mathcal{O}(m/\log(1/\epsilon))$, the number of without-replacement stochastic iterations is smaller than the data size $m$. Thus, assuming the data is randomly shuffled, we can get a solution using only *sequential* data passes, without the need to reshuffle.

In terms of the log factors, we note that the condition $\max_t F(\mathbf{w}_t) - F(\mathbf{w}^*) \leq B$ with probability 1 is needed for technical reasons in our analysis, and we conjecture that it can be improved. However, since $B$ appears only inside log factors, even a crude bound would suffice. In appendix C, we indeed show that under there is always a valid $B$ satisfying $\log(B) = \mathcal{O}(\log(1/\epsilon)\log(T) + \log(1/\lambda))$. Regarding the logarithmic dependence on the dimension $d$, it is due to an application of a matrix Bernstein inequality for $d \times d$ matrices, and can possibly be improved.

## 5.1 Application of Without-Replacement SVRG to distributed learning

An important variant of the problems we discussed so far is when training data (or equivalently, the individual loss functions $f_1(\cdot), \ldots, f_m(\cdot)$) are split between different machines, who need to communicate in order to reach a good solution. This naturally models situations where data is too large to fit at a single machine, or where we wish to speed up the optimization process by parallelizing it on several computers. Over the past few years, there has been much research on this question in the machine learning community, with just a few examples including [24, 2, 1, 5, 4, 10, 20, 19, 25, 13].

A substantial number of these works focus on the setting where the data is split equally *at random* between $k$ machines (or equivalently, that data is assigned to each machine by sampling without replacement from $\{f_1(\cdot), \ldots, f_m(\cdot)\}$)). Intuitively, this creates statistical similarities between the data at different machines, which can be leveraged to aid the optimization process. Recently, Lee et al. [13] proposed a simple and elegant approach, which applies at least in certain parameter regimes. This is based on the observation that SVRG, according to its with-replacement analysis, requires $\mathcal{O}(\log(1/\epsilon))$ epochs, where in each epoch one needs to compute an exact gradient of the objective function $F(\cdot) = \frac{1}{m} \sum_{i=1}^{m} f_i(\cdot)$, and $\mathcal{O}(1/\lambda)$ gradients of individual losses $f_i(\cdot)$ chosen uniformly at random (where $\epsilon$ is the required suboptimality, and $\lambda$ is the strong convexity parameter of $F(\cdot)$). Therefore, if each machine had *i.i.d.* samples from $\{f_1(\cdot), \ldots, f_m(\cdot)\}$, whose union cover $\{f_1(\cdot), \ldots, f_m(\cdot)\}$, the machines could just simulate SVRG: First, each machine splits its data to batches of size $\mathcal{O}(1/\lambda)$. Then, each SVRG epoch is simulated by the machines computing a gradient of $F(\cdot) = \frac{1}{m} \sum_{i=1}^{m} f_i(\cdot)$ – which can be fully parallelized and involves one communication round (assuming a broadcast communication model) – and one machine computing gradients with respect to one of its batches. Overall, this would require $\mathcal{O}(\log(1/\epsilon))$ communication rounds, and $\mathcal{O}(m/k + 1/\lambda) \log(1/\epsilon)$ runtime, where $m/k$ is the number of data points per machine (ignoring communication time, and assuming constant time to compute a gradient of $f_i(\cdot)$). Under the reasonable assumption that the strong convexity parameter $\lambda$ is at least $k/m$, this requires runtime *linear* in the data size per machine, and *logarithmic* in the target accuracy $\epsilon$, with only a *logarithmic* number of communication rounds. Up to log factors, this is essentially the best one can hope for with a distributed algorithm. Moreover, a lower bound in [13] indicates that at least in the worst case, $\mathcal{O}(\log(1/\epsilon))$ communication rounds is impossible to obtain if $\lambda$ is significantly smaller than $k/m$.

Unfortunately, the reasoning above crucially relies on each machine having access to i.i.d. samples, which can be reasonable in some cases, but is different than the more common assumption that the data is randomly *split* among the machines. To circumvent this issue, [13] propose to communicate individual data points / losses $f_i(\cdot)$ between machines, so as to simulate i.i.d. sampling. However, by the birthday paradox, this only works well in the regime where the overall number of samples required ($\mathcal{O}((1/\lambda) \log(1/\epsilon))$) is not much larger than $\sqrt{m}$. Otherwise, with-replacement and without-replacement sampling becomes statistically very different, and a large number of data points would need to be communicated. In other words, if communication is an expensive resource, then the solution proposed in [13] only works well when $\lambda$ is at least order of $1/\sqrt{m}$. In machine learning applications, the strong convexity parameter $\lambda$ often comes from explicit regularization designed to prevent over-fitting, and needs to scale with the data size, usually between $1/\sqrt{m}$ and $1/m$. Thus, the solution above is communication-efficient only when $\lambda$ is relatively large.

However, the situation immediately improves if we can use a *without-replacement* version of SVRG, which can easily be simulated with randomly partitioned data: The stochastic batches can now be simply subsets of each machine's data, which are statistically identical to sampling $\{f_1(\cdot), \ldots, f_m(\cdot)\}$ without replacement. Thus, no data points need to be sent across machines, even if $\lambda$ is small. For clarity, we present an explicit pseudocode as Algorithm 2 in Appendix D.

Let us consider the analysis of no-replacement SVRG provided in Thm. 4. According to this analysis, by setting $T = \Theta(1/\lambda)$, then as long as the total number of batches is at least $\Omega(\log(1/\epsilon))$, and $\lambda = \tilde{\Omega}(1/m)$, then the algorithm will attain an $\epsilon$-suboptimal solution in expectation. In other words, without any additional communication, we extend the applicability of distributed SVRG (at least for regularized least squares) from the $\lambda = \tilde{\Omega}(1/\sqrt{m})$ regime to $1/\lambda = \tilde{\Omega}(1/m)$.

We emphasize that this formal analysis only applies to regularized squared loss, which is the scope of Thm. 4. However, this algorithmic approach can be applied to any loss function, and we conjecture that it will have similar performance for any smooth losses and strongly-convex objectives.

**Acknowledgments:** This research is supported in part by an FP7 Marie Curie CIG grant, an ISF grant 425/13, and by the Intel Collaborative Research Institute for Computational Intelligence (ICRI-CI).

## Footnotes

[1]For simplicity, we assume the algorithm is deterministic given $f_1, \ldots, f_m$, but all results in this section also hold for randomized algorithms (in expectation over the algorithm's randomness), assuming the *expected* regret of the algorithm w.r.t. any $\mathbf{w} \in \mathcal{W}$ is at most $R_T$.

[2]$G$ is generally on the order of $L + \lambda B$, which is the same as $L + \mu B$ if $L$ is the dominant term. This happens for instance with the squared loss, whose Lipschitz parameter is on the order of $\mu B$.

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
