[Supplementary Material · repSGD_final_supplementary.pdf]

# A  Proofs

Before providing the proofs of our main theorems, we develop in Subsection A.1 below some important technical results on transductive Rademacher complexity, required in some of the proofs.

## A.1  Results on Transductive Rademacher Complexity

In this subsection, we develop a few important tools and result about Rademacher complexity, that will come handy in our analysis. We begin by the following theorem, which is a straightforward corollary of Theorem 1 from [6] (attained by simplifying and upper-bounding some of the parameters).

**Theorem 5.** *Suppose $\mathcal{V} \subseteq [-B, B]^m$ for some $B > 0$. Let $\sigma$ be a permutation over $\{1, \dots, m\}$ chosen uniformly at random, and define $\mathbf{v}_{1:s} = \frac{1}{s} \sum_{j=1}^{s} v_{\sigma(j)}$, $\mathbf{v}_{s+1:m} = \frac{1}{u} \sum_{j=s+1}^{m} v_{\sigma(j)}$. Then for any $\delta \in (0, 1)$, with probability at least $1 - \delta$,*

$$\sup_{\mathbf{v} \in \mathcal{V}} (\mathbf{v}_{1:s} - \mathbf{v}_{s+1:m}) \leq \mathcal{R}_{s,u}(\mathcal{V}) + 6B \left( \frac{1}{\sqrt{s}} + \frac{1}{\sqrt{u}} \right) \left( 1 + \log \left( \frac{1}{\delta} \right) \right).$$

We note that the theorem in [6] actually bounds $\mathbf{v}_{s+1:m} - \mathbf{v}_{1:s}$, but the proof is completely symmetric to the roles of $\mathbf{v}_{s+1:m}$ and $\mathbf{v}_{1:s}$, and hence also implies the formulation above.

We will also need the well-known contraction property, which states that the Rademacher complexity of a class of vectors $\mathcal{V}$ can only increase by a factor of $L$ if we apply on each coordinate a fixed $L$-Lipschitz function:

**Lemma 2.** *Let $g_1, \dots, g_m$ are real-valued, $L$-Lipschitz functions, and given some $\mathcal{V} \subseteq \mathbb{R}^m$, define $\mathbf{g} \circ \mathcal{V} = \{(g_1(v_1), \dots, g_m(v_m)) : (v_1, \dots, v_m) \in \mathcal{V}\}$. Then*

$$\mathcal{R}_{s,u}(\mathbf{g} \circ \mathcal{V}) \leq L \cdot \mathcal{R}_{s,u}(\mathcal{V}).$$

This is a slight generalization of Lemma 5 from [6] (which is stated for $g_1 = g_2 = \dots = g_m$, but the proof is exactly the same).

In our analysis, we will actually only need bounds on the expectation of $\mathbf{v}_{1:s} - \mathbf{v}_{s+1:m}$, which is weaker than what Thm. 5 provides. Although such a bound can be developed from scratch, we find it more convenient to simply get such a bound from Thm. 5. Specifically, combining Thm. 5 and Lemma 7 from Appendix B, we have the following straightforward corollary:

**Corollary 2.** *Suppose $\mathcal{V} \subseteq [-B, B]^m$ for some $B > 0$. Let $\sigma$ be a permutation over $\{1, \dots, m\}$ chosen uniformly at random, and define $\mathbf{v}_{1:t-1} = \frac{1}{t-1} \sum_{j=1}^{t-1} v_{\sigma(j)}$, $\mathbf{v}_{t:m} = \frac{1}{m-t+1} \sum_{j=t}^{m} v_{\sigma(j)}$. Then*

$$\mathbb{E} \left[ \sup_{\mathbf{v} \in \mathcal{V}} \mathbf{v}_{1:t-1} - \mathbf{v}_{t:m} \right] \leq \mathcal{R}_{t-1, m-t+1}(\mathcal{V}) + 12B \left( \frac{1}{\sqrt{t-1}} + \frac{1}{\sqrt{m-t+1}} \right).$$

*Moreover, if $\sup_{\mathbf{v} \in \mathcal{V}} (\mathbf{v}_{1:t-1} - \mathbf{v}_{t:m}) \geq 0$ for any permutation $\sigma$, then*

$$\sqrt{\mathbb{E} \left[ \left( \sup_{\mathbf{v} \in \mathcal{V}} \mathbf{v}_{1:t-1} - \mathbf{v}_{t:m} \right)^2 \right]} \leq \sqrt{2} \cdot \mathcal{R}_{s,u}(\mathcal{V}) + 12\sqrt{2}B \left( \frac{1}{\sqrt{t-1}} + \frac{1}{\sqrt{m-t+1}} \right).$$

We now turn to collect a few other structural results, which will be useful when studying the Rademacher complexity of linear predictors or loss gradients derived from such predictors.

**Lemma 3.** *Given two sets of vectors $\mathcal{V} \in [-B_{\mathcal{V}}, B_{\mathcal{V}}]^m$, $\mathcal{S} \subseteq [-B_{\mathcal{S}}, B_{\mathcal{S}}]^m$ for some $B_{\mathcal{V}}, B_{\mathcal{S}} \geq 0$, define*

$$\mathcal{U} = \{(v_1 s_1, \dots, v_m s_m) : (v_1, \dots, v_m) \in \mathcal{V}, (s_1, \dots, s_m) \in \mathcal{S}\}.$$

*Then*

$$\mathcal{R}_{s,u}(\mathcal{U}) \leq B_{\mathcal{S}} \cdot \mathcal{R}_{s,u}(\mathcal{V}) + B_{\mathcal{V}} \cdot \mathcal{R}_{s,u}(\mathcal{S}).$$

*Proof.* The proof resembles the proof of the contraction inequality for standard Rademacher complexity (see for instance Lemma 26.9 in [18]).

By definition of $\mathcal{R}_{s,u}$, it is enough to prove that

$$\mathbb{E}_{r_1,\ldots,r_m}\left[\sup_{\mathbf{v},\mathbf{s}}\sum_{i=1}^{m}r_i v_i s_i\right] \leq \mathbb{E}_{r_1,\ldots,r_m}\left[\sup_{\mathbf{v},\mathbf{s}}\sum_{i=1}^{m}r_i(B_{\mathcal{S}}\cdot v_i + B_{\mathcal{V}}\cdot s_i)\right], \tag{3}$$

since the right hand side can be upper bounded by $B_{\mathcal{S}}\cdot\mathbb{E}[\sup_{\mathbf{v}}\sum_{i=1}^{m}r_i v_i] + B_{\mathcal{V}}\cdot\mathbb{E}[\sup_{\mathbf{s}}\sum_{i=1}^{m}r_i s_i]$. To get this, we will treat the coordinates one-by-one, starting with the first coordinate and showing that

$$\mathbb{E}_{r_1,\ldots,r_m}\left[\sup_{\mathbf{v},\mathbf{s}}\sum_{i=1}^{m}r_i v_i s_i\right] \leq \mathbb{E}_{r_1,\ldots,r_m}\left[\sup_{\mathbf{v},\mathbf{s}}\left(r_1(B_{\mathcal{S}}\cdot v_1 + B_{\mathcal{V}}\cdot s_1) + \sum_{i=2}^{m}r_i v_i s_i\right)\right]. \tag{4}$$

Repeating the same argument for coordinates $2, 3, \ldots, m$ will yield Eq. (3).

For any values $v, v'$ and $s, s'$ in the coordinates of some $\mathbf{v}\in\mathcal{V}$ and $\mathbf{s}\in\mathcal{S}$ respectively, we have

$$\begin{aligned}|vs - v's'| &= |vs - v's + v's - v's'| \leq |vs - v's| + |v's - v's'| \\ &\leq |v - v'|\cdot|s| + |s - s'|\cdot|v| \leq B_{\mathcal{S}}|v - v'| + B_{\mathcal{V}}|s - s'|.\end{aligned} \tag{5}$$

Recalling that $r_i$ are i.i.d. and take values of $+1$ and $-1$ with probability $p$ (and $0$ otherwise), we can write the left hand side of Eq. (4) as

$$\begin{aligned}\mathbb{E}_{r_1,\ldots,r_m}&\left[\sup_{\mathbf{v},\mathbf{s}}\left(r_1 v_1 s_1 + \sum_{i=2}^{m}r_i v_i s_i\right)\right] \\ =\mathbb{E}_{r_2,\ldots,r_m}&\left[p\cdot\sup_{\mathbf{v},\mathbf{s}}\left(v_1 s_1 + \sum_{i=2}^{m}r_i v_i s_i\right) + p\cdot\sup_{\mathbf{v},\mathbf{s}}\left(-v_1 s_1 + \sum_{i=2}^{m}r_i v_i s_i\right)\right. \\ &\left.+(1-2p)\sup_{\mathbf{v},\mathbf{s}}\left(\sum_{i=2}^{m}r_i v_i s_i\right)\right] \\ =\mathbb{E}_{r_2,\ldots,r_m}&\left[\sup_{\mathbf{v},\mathbf{s}}\left(p\,v_1 s_1 + p\sum_{i=2}^{m}r_i v_i s_i\right) + \sup_{\mathbf{v}',\mathbf{s}'}\left(-p\,v_1 s_1 + p\sum_{i=2}^{m}r_i v_i' s_i'\right)\right. \\ &\left.+\sup_{\mathbf{v}'',\mathbf{s}''}\left((1-2p)\sum_{i=2}^{m}r_i v_i'' s_i''\right)\right] \\ =\mathbb{E}_{r_2,\ldots,r_m}&\left[\sup_{\mathbf{v},\mathbf{v}',\mathbf{v}'',\mathbf{s},\mathbf{s}'\mathbf{s}''}\left(p(v_1 s_1 - v_1' s_1') + p\sum_{i=2}^{m}r_i v_i s_i + p\sum_{i=2}^{m}r_i v_i' s_i'\right.\right. \\ &\left.\left.+(1-2p)\sum_{i=2}^{m}r_i v_i'' s_i''\right)\right].\end{aligned}$$

Using Eq. (5) and the fact that we take a supremum over $\mathbf{v}$, $\mathbf{v}'$ and $\mathbf{s}$, $\mathbf{s}'$ from the same sets, this equals

$$= \mathbb{E}_{r_2,\ldots,r_m} \left[ \sup_{\mathbf{v},\mathbf{v}',\mathbf{v}'',\mathbf{s},\mathbf{s}'\mathbf{s}''} \left( p\left(B_{\mathcal{S}}|v_1 - v_1'| + B_{\mathcal{V}}|s_1 - s_1'|\right) + p\sum_{i=2}^{m} r_i v_i s_i + p\sum_{i=2}^{m} r_i v_i' s_i' \right. \right.$$
$$\left. \left. + (1 - 2p)\sum_{i=2}^{m} r_i v_i'' s_i'' \right) \right]$$

$$= \mathbb{E}_{r_2,\ldots,r_m} \left[ \sup_{\mathbf{v},\mathbf{v}',\mathbf{v}'',\mathbf{s},\mathbf{s}'\mathbf{s}''} \left( p\left(B_{\mathcal{S}}(v_1 - v_1') + B_{\mathcal{V}}(s_1 - s_1')\right) + p\sum_{i=2}^{m} r_i v_i s_i + p\sum_{i=2}^{m} r_i v_i' s_i' \right. \right.$$
$$\left. \left. + (1 - 2p)\sum_{i=2}^{m} r_i v_i'' s_i'' \right) \right]$$

$$= \mathbb{E}_{r_2,\ldots,r_m} \left[ p \sup_{\mathbf{v},\mathbf{s}} \left( (B_{\mathcal{S}} \cdot v_1 + B_{\mathcal{V}} \cdot s_1) + \sum_{i=2}^{m} r_i v_i s_i \right) + p \sup_{\mathbf{v}',\mathbf{s}'} \left( p(-B_{\mathcal{S}} v_1' - B_{\mathcal{V}} s_1') + p\sum_{i=2}^{m} r_i v_i' s_i' \right) \right.$$
$$\left. + (1 - 2p) \sup_{\mathbf{v},\mathbf{s}''} \left( \sum_{i=2}^{m} r_i v_i'' s_i'' \right) \right]$$

$$= \mathbb{E}_{r_1,\ldots,r_m} \left[ \sup_{\mathbf{v},\mathbf{s}} \left( r_1(B_{\mathcal{S}} \cdot v_1 + B_{\mathcal{V}} \cdot s_1) + \sum_{i=2}^{m} r_i v_i s_i \right) \right]$$

as required. $\qquad\square$

**Lemma 4.** *For some $B > 0$ and vectors $\mathbf{x}_1, \ldots, \mathbf{x}_m$ with Euclidean norm at most 1, let*

$$\mathcal{V}_B = \{(\langle \mathbf{w}, \mathbf{x}_1\rangle, \ldots, \langle \mathbf{w}, \mathbf{x}_m\rangle) : \|\mathbf{w}\| \leq B\}.$$

*Then*

$$\mathcal{R}_{s,u}(\mathcal{V}_B) \leq \sqrt{2}B\left(\frac{1}{\sqrt{s}} + \frac{1}{\sqrt{u}}\right).$$

*Proof.* Using the definition of $\mathcal{V}_B$ and applying Cauchy-Schwartz,

$$\mathbb{E}_{r_1,\ldots,r_m}\left[\sup_{\mathbf{v}\in\mathcal{V}}\sum_{i=1}^{m} r_i v_i\right] = \mathbb{E}_{r_1,\ldots,r_m}\left[\sup_{\mathbf{w}:\|\mathbf{w}\|\leq B}\left\langle \mathbf{w}, \sum_{i=1}^{m} r_i \mathbf{x}_i\right\rangle\right] \leq \mathbb{E}_{r_1,\ldots,r_m}\left[B\left\|\sum_{i=1}^{m} r_i \mathbf{x}_i\right\|\right]$$

$$\leq B\sqrt{\mathbb{E}_{r_1,\ldots,r_m}\left[\left\|\sum_{i=1}^{m} r_i \mathbf{x}_i\right\|^2\right]} = B\sqrt{\mathbb{E}_{r_1,\ldots,r_m}\sum_{i,j=1}^{m} r_i r_j \langle \mathbf{x}_i, \mathbf{x}_j\rangle}.$$

Recall that $r_i$ are independent and equal $+1, -1$ with probability $p$ (and 0 otherwise). Therefore, for $i \neq j$, $\mathbb{E}[r_i r_j] = 0$, and if $i = j$, $\mathbb{E}[r_i r_j] = \mathbb{E}[r_i^2] = 2p$. Using this and the assumption that $\|\mathbf{x}_i\| \leq 1$ for all $i$, the above equals

$$B\sqrt{\sum_{i=1}^{m} 2p \langle \mathbf{x}_i, \mathbf{x}_i\rangle} \leq B\sqrt{2pm}.$$

Therefore, $\mathcal{R}_{s,u}(\mathcal{V}_B) \leq \left(\frac{1}{s} + \frac{1}{u}\right) B\sqrt{2pm}$. Recalling that $p = \frac{su}{(s+u)^2}$ where $s + u = m$ and plugging it in, we get the upper bound

$$B\left(\frac{1}{s} + \frac{1}{u}\right)\sqrt{2\frac{su}{(s+u)^2}(s+u)} = \sqrt{2}B\left(\frac{1}{s} + \frac{1}{u}\right)\sqrt{\frac{su}{s+u}}$$

$$= \sqrt{2}B\left(\frac{1}{s}\sqrt{s\frac{u}{s+u}} + \frac{1}{u}\sqrt{u\frac{s}{s+u}}\right) \leq \sqrt{2}B\left(\frac{1}{\sqrt{s}} + \frac{1}{\sqrt{u}}\right),$$

from which the result follows. $\qquad\square$

Combining Corollary 2, Lemma 2, Lemma 3 and Lemma 4, we have the following:

**Corollary 3.** *Suppose the functions $f_1(\cdot), \ldots, f_m(\cdot)$ are of the form $f_i(\mathbf{w}) = \ell_i(\langle \mathbf{w}, \mathbf{x}_i \rangle) + r(\mathbf{w})$, where $\|\mathbf{x}_i\| \leq 1$ and $r$ is some fixed function. Let $B > 0$ such that the iterates $\mathbf{w}$ chosen by the algorithm are from a set $\mathcal{W} \subseteq \{\mathbf{w} : \|\mathbf{w}\| \leq B\}$ which contains the origin $\mathbf{0}$. Finally, assume $\ell_i(\cdot)$ is $L$-Lipschitz and $\mu$-smooth over the interval $[-B, B]$. Then*

$$\sqrt{\mathbb{E}\left[\left(\sup_{\mathbf{w} \in \mathcal{W}} \left\langle \nabla F_{1:t-1}(\mathbf{w}) - \nabla F_{t:m}(\mathbf{w}), \frac{\mathbf{w}}{\|\mathbf{w}\|} \right\rangle\right)^2\right]}$$

$$\leq (19L + 2\mu B)\left(\frac{1}{\sqrt{t-1}} + \frac{1}{\sqrt{m-t+1}}\right).$$

*Proof.* By definition of $f_i$, we have

$$\left\langle \nabla f_i(\mathbf{w}), \frac{\mathbf{w}}{\|\mathbf{w}\|} \right\rangle = \ell_i'(\langle \mathbf{w}, \mathbf{x}_i \rangle)\left\langle \frac{\mathbf{w}}{\|\mathbf{w}\|}, \mathbf{x}_i \right\rangle + \left\langle \nabla r(\mathbf{w}), \frac{\mathbf{w}}{\|\mathbf{w}\|} \right\rangle.$$

Therefore, the expression in the corollary statement can be written as

$$\sqrt{\mathbb{E}\left[\left(\sup_{\mathbf{u} \in \mathcal{U}} u_{1:t-1} - u_{t:m}\right)^2\right]},$$

where

$$u_i = \ell_i'(\langle \mathbf{w}, \mathbf{x}_i \rangle)\left\langle \frac{\mathbf{w}}{\|\mathbf{w}\|}, \mathbf{x}_i \right\rangle$$

and

$$\mathcal{U} = \left\{\left(\ell_1'(\langle \mathbf{w}, \mathbf{x}_1 \rangle)\left\langle \frac{\mathbf{w}}{\|\mathbf{w}\|}, \mathbf{x}_1 \right\rangle, \ldots, \ell_m'(\langle \mathbf{w}, \mathbf{x}_m \rangle)\left\langle \frac{\mathbf{w}}{\|\mathbf{w}\|}, \mathbf{x}_m \right\rangle\right) : \|\mathbf{w}\| \in \mathcal{W}\right\}$$

(note that the terms involving $r$ get cancelled out in the expression $u_{1:s} - u_{s+1:m}$, so we may drop them). Applying Corollary 2 (noting that $|u_i| \leq L$, and that $\sup_{\mathbf{u} \in \mathcal{U}} u_{1:t-1} - u_{t:m} \geq 0$, since $u_{1:t-1} - u_{t:m} = 0$ if we choose $\mathbf{w} = \mathbf{0}$), we have

$$\sqrt{\mathbb{E}\left[\left(\sup_{\mathbf{u} \in \mathcal{U}} u_{1:t-1} - u_{t:m}\right)^2\right]} \leq \sqrt{2} \cdot \mathcal{R}_{t-1,m-t+1}(\mathcal{U}) + 12\sqrt{2}L\left(\frac{1}{\sqrt{t-1}} + \frac{1}{\sqrt{m-t+1}}\right). \tag{6}$$

Now, define

$$\mathcal{V} = \{(\ell_1'(\langle \mathbf{w}, \mathbf{x}_1 \rangle), \ldots, \ell_m'(\langle \mathbf{w}, \mathbf{x}_m \rangle)) : \|\mathbf{w}\| \leq B\}$$

and

$$\mathcal{S} = \{(\langle \mathbf{w}, \mathbf{x}_1 \rangle, \ldots, \langle \mathbf{w}, \mathbf{x}_m \rangle) : \|\mathbf{w}\| = 1\},$$

and note that $\mathcal{U}$ as we defined it satisfies

$$\mathcal{U} \subseteq \{(\ell_1'(v_1)s_1, \ldots, \ell_m'(v_m)s_m) : (v_1, \ldots, v_m) \in \mathcal{V}, (s_1, \ldots, s_m) \in \mathcal{S}\}.$$

Moreover, by construction, the coordinates of each $\mathbf{v} \in \mathcal{V}$ are bounded in $[-L, L]$, and the coordinates of each $\mathbf{s} \in \mathcal{S}$ are bounded in $[-1, +1]$. Applying Lemma 3, we get

$$\mathcal{R}_{t-1,m-t+1}(\mathcal{U}) \leq \mathcal{R}_{t-1,m-t+1}(\mathcal{V}) + L \cdot \mathcal{R}_{t-1,m-t+1}(\mathcal{S}). \tag{7}$$

Using Lemma 4, we have

$$\mathcal{R}_{t-1,m-t+1}(\mathcal{S}) \leq \sqrt{2}\left(\frac{1}{\sqrt{t-1}} + \frac{1}{\sqrt{m-t+1}}\right). \tag{8}$$

Finally, applying Lemma 2 (using the fact that each $\ell_i'$ is $\mu$-Lipschitz) followed by Lemma 4, we have

$$\mathcal{R}_{t-1,m-t+1}(\mathcal{V}) \leq \sqrt{2}\mu B\left(\frac{1}{\sqrt{t-1}} + \frac{1}{\sqrt{m-t+1}}\right). \tag{9}$$

Combining Eq. (7), Eq. (8) and Eq. (9), plugging into Eq. (6), and slightly simplifying for readability, yields the desired result. $\square$

## A.2 Proof of Lemma 1

The lemma is immediate when $t = 1$, so we will assume $t > 1$. Also, we will prove it when the expectation $\mathbb{E}$ is conditioned on $\sigma(1), \ldots, \sigma(t-1)$, and the result will follow by taking expectations over them. With this conditioning, $s_1, \ldots, s_m$ have some fixed distribution, which is independent of how $\sigma$ permutes $\{1, \ldots, m\} \setminus \{\sigma(1), \ldots, \sigma(t-1)\}$.

Recall that $\sigma$ is chosen uniformly at random. Therefore, conditioned on $\sigma(1), \ldots, \sigma(t-1)$, the value of $\sigma(t)$ is uniformly distributed on $\{1, \ldots, m\} \setminus \{\sigma(1), \ldots, \sigma(t-1)\}$, which is the same set as $\sigma(t), \ldots, \sigma(m)$. Therefore, the left hand side in the lemma statement equals

$$
\mathbb{E}\left[\frac{1}{m}\sum_{i=1}^{m} s_i - \frac{1}{m-t+1}\sum_{i=t}^{m} s_{\sigma(i)}\right]
$$

$$
= \mathbb{E}\left[\frac{1}{m}\sum_{i=1}^{m} s_{\sigma(i)} - \frac{1}{m-t+1}\sum_{i=t}^{m} s_{\sigma(i)}\right]
$$

$$
= \mathbb{E}\left[\frac{1}{m}\sum_{i=1}^{t-1} s_{\sigma(i)} + \left(\frac{1}{m} - \frac{1}{m-t+1}\right)\sum_{i=t}^{m} s_{\sigma(i)}\right]
$$

$$
= \mathbb{E}\left[\frac{1}{m}\sum_{i=1}^{t-1} s_{\sigma(i)} - \frac{t-1}{m(m-t+1)}\sum_{i=t}^{m} s_{\sigma(i)}\right]
$$

$$
= \frac{t-1}{m} \cdot \mathbb{E}\left[\frac{1}{t-1}\sum_{i=1}^{t-1} s_{\sigma(i)} - \frac{1}{m-t+1}\sum_{i=t}^{m} s_{\sigma(i)}\right]
$$

as required.

## A.3 Proof of Thm. 2

Let $\mathcal{V} = \{(f_1(\mathbf{w}), \ldots, f_m(\mathbf{w})) \mid \mathbf{w} \in \mathcal{W}\}$ and applying Corollary 2, we have

$$
\mathbb{E}[F_{1:t-1}(\mathbf{w}_t) - F_{t:m}(\mathbf{w}_t)] \leq \mathbb{E}\left[\sup_{\mathbf{w}\in\mathcal{W}} F_{1:t-1}(\mathbf{w}) - F_{t:m}(\mathbf{w})\right]
$$

$$
\leq \mathcal{R}_{t-1:m-t+1}(\mathcal{V}) + 12B\left(\frac{1}{\sqrt{t-1}} + \frac{1}{\sqrt{m-t+1}}\right).
$$

Plugging this into the bound from Thm. 1, we have

$$
\mathbb{E}\left[F(\bar{\mathbf{w}}_T) - F(\mathbf{w}^*)\right] \leq \frac{R_T}{T} + \frac{1}{mT}\sum_{t=2}^{T}(t-1)\left(\mathcal{R}_{t-1:m-t+1}(\mathcal{V}) + 12B\left(\frac{1}{\sqrt{t-1}} + \frac{1}{\sqrt{m-t+1}}\right)\right)
$$

Applying Lemma 6, the right hand side is at most

$$
\frac{R_T}{T} + \frac{1}{mT}\sum_{t=2}^{T}(t-1)\mathcal{R}_{t-1:m-t+1}(\mathcal{V}) + \frac{24B}{\sqrt{m}}. \tag{10}
$$

## A.4 Proof of Corollary 1

Note that all terms in the bound of Thm. 2, except the regret term, are obtained by considering the difference $F_{1:t-1}(\mathbf{w}_t) - F_{t:m}(\mathbf{w}_t)$, so any additive terms in the losses which are constant (independent of $i$) are cancelled out. Therefore, we may assume without loss of generality that $f_i(\mathbf{w}) = \ell_i(\langle \mathbf{w}, \mathbf{x}_i \rangle)$ (without the $r(\mathbf{w})$ term), and that $\ell_i$ is centered so that $\ell_i(0) = 0$. Applying Lemma 2 and Lemma 4, we can upper the Rademacher complexity as follows:

$$
\mathcal{R}_{t-1:m-t+1}(\mathcal{V}) \leq L \cdot \mathcal{R}_{t-1:m-t+1}\left(\{(\langle \mathbf{w}, \mathbf{x}_1 \rangle, \ldots, \langle \mathbf{w}, \mathbf{x}_m \rangle) \mid \mathbf{w} \in \mathcal{W}\}\right)
$$

$$
\leq \sqrt{2} \cdot BL\left(\frac{1}{\sqrt{t-1}} + \frac{1}{\sqrt{m-t+1}}\right).
$$

Plugging this into Thm. 2, applying Lemma 6, and noting that by the assumptions above and in the corollary statement, $\sup_{i,\mathbf{w}\in\mathcal{W}} |f_i(\mathbf{w})| \leq \sup_{a\in[-\bar{B},\bar{B}]} |\ell_i(a)| \leq \bar{B}L$, the result follows.

## A.5 Proof of Thm. 3

Since the algorithm is invariant to shifting the coordinates or shifting all loss functions by a constant, we will assume without loss of generality that $\mathcal{W}$ contains the origin $\mathbf{0}$ (and therefore $\mathcal{W} \subseteq \{\mathbf{w} : \|\mathbf{w}\| \leq B\}$), that the objective function $F(\cdot)$ is minimized at $\mathbf{0}$, and that $F(\mathbf{0}) = 0$. By definition of the algorithm and convexity of $\mathcal{W}$, we have

$$\mathbb{E}[\|\mathbf{w}_{t+1}\|^2] = \mathbb{E}\left[\left\|\Pi_{\mathcal{W}}(\mathbf{w}_t - \eta_t \nabla f_{\sigma(t)}(\mathbf{w}_t))\right\|^2\right] \leq \mathbb{E}\left[\left\|\mathbf{w}_t - \eta_t \nabla f_{\sigma(t)}(\mathbf{w}_t)\right\|^2\right]$$

$$\leq \mathbb{E}\left[\|\mathbf{w}_t\|^2\right] - 2\eta_t \mathbb{E}\left[\langle \nabla f_{\sigma(t)}(\mathbf{w}_t), \mathbf{w}_t\rangle\right] + \eta_t^2 G^2$$

$$= \mathbb{E}\left[\|\mathbf{w}_t\|^2\right] - 2\eta_t \mathbb{E}\left[\langle \nabla F(\mathbf{w}_t), \mathbf{w}_t\rangle\right] + 2\eta_t \mathbb{E}\left[\langle \nabla F(\mathbf{w}_t) - \nabla f_{\sigma(t)}(\mathbf{w}_t), \mathbf{w}_t\rangle\right] + \eta_t^2 G^2.$$

By definition of strong convexity, since $F(\cdot)$ is $\lambda$-strongly convex, minimized at $\mathbf{0}$, and assumed to equal 0 there, we have $\langle \nabla F(\mathbf{w}_t), \mathbf{w}_t\rangle \geq F(\mathbf{w}_t) + \frac{\lambda}{2}\|\mathbf{w}\|^2$. Plugging this in, changing sides and dividing by $2\eta_t$, we get

$$\mathbb{E}[F(\mathbf{w}_t)] \leq \left(\frac{1}{2\eta_t} - \frac{\lambda}{2}\right)\mathbb{E}[\|\mathbf{w}_t\|^2] - \frac{1}{2\eta_t}\cdot\mathbb{E}[\|\mathbf{w}_{t+1}\|^2] + \mathbb{E}\left[\langle \nabla F(\mathbf{w}_t) - \nabla f_{\sigma(t)}(\mathbf{w}_t), \mathbf{w}_t\rangle\right] + \frac{\eta_t}{2}G^2. \tag{11}$$

We now turn to treat the third term in the right hand side above. Since $\mathbf{w}_t$ (as a random variable over the permutation $\sigma$ of the data) depends only on $\sigma(1), \ldots, \sigma(t-1)$, we can use Lemma 1 and Cauchy-Schwartz, to get

$$\mathbb{E}\left[\langle \nabla F(\mathbf{w}_t) - \nabla f_{\sigma(t)}(\mathbf{w}_t), \mathbf{w}_t\rangle\right] = \mathbb{E}\left[\left\langle \frac{1}{m}\sum_{i=1}^m \nabla f_i(\mathbf{w}_t) - \nabla f_{\sigma(t)}(\mathbf{w}_t), \mathbf{w}_t\right\rangle\right]$$

$$= \frac{t-1}{m}\cdot\mathbb{E}\left[\left(\langle\nabla F_{1:t-1}(\mathbf{w}_t) - \nabla F_{t:m}(\mathbf{w}_t), \mathbf{w}_t\rangle\right)\right]$$

$$= \frac{t-1}{m}\cdot\mathbb{E}\left[\|\mathbf{w}_t\|\cdot\left\langle\nabla F_{1:t-1}(\mathbf{w}_t) - \nabla F_{t:m}(\mathbf{w}_t), \frac{\mathbf{w}_t}{\|\mathbf{w}_t\|}\right\rangle\right]$$

$$\leq \frac{t-1}{m}\cdot\mathbb{E}\left[\|\mathbf{w}_t\|\cdot\sup_{\mathbf{w}\in\mathcal{W}}\left\langle\nabla F_{1:t-1}(\mathbf{w}) - \nabla F_{t:m}(\mathbf{w}), \frac{\mathbf{w}}{\|\mathbf{w}\|}\right\rangle\right]$$

$$\leq \frac{t-1}{m}\cdot\sqrt{\mathbb{E}\left[\|\mathbf{w}_t\|^2\right]}\cdot\sqrt{\mathbb{E}\left[\left(\sup_{\mathbf{w}\in\mathcal{W}}\left\langle\nabla F_{1:t-1}(\mathbf{w}) - \nabla F_{t:m}(\mathbf{w}), \frac{\mathbf{w}}{\|\mathbf{w}\|}\right\rangle\right)^2\right]}$$

Applying Corollary 3 (using the convention $0/\sqrt{0} = 0$ in the case $t = 1$ where the expression above is 0 anyway), this is at most

$$\frac{t-1}{m}\cdot\sqrt{\mathbb{E}\left[\|\mathbf{w}_t\|^2\right]}\cdot(19L + 2\mu B)\left(\frac{1}{\sqrt{t-1}} + \frac{1}{\sqrt{m-t+1}}\right)$$

$$= \sqrt{\mathbb{E}\left[\|\mathbf{w}_t\|^2\right]}\cdot\frac{19L + 2\mu B}{m}\left(\sqrt{t-1} + \frac{t-1}{\sqrt{m-t+1}}\right).$$

Using the fact that for any $a, b \geq 0$, $\sqrt{ab} = \sqrt{\frac{\lambda}{2}a\cdot\frac{2}{\lambda}b} \leq \frac{\lambda}{4}a + \frac{1}{\lambda}b$ by the arithmetic-geometric mean inequality, the above is at most

$$\frac{\lambda}{4}\cdot\mathbb{E}[\|\mathbf{w}_t\|^2] + \frac{(19L + 2\mu b)^2}{\lambda m^2}\left(\sqrt{t-1} + \frac{t-1}{\sqrt{m-t+1}}\right)^2.$$

Since $(a+b)^2 \leq 2(a^2 + b^2)$, this is at most

$$\frac{\lambda}{4}\cdot\mathbb{E}[\|\mathbf{w}_t\|^2] + \frac{2(19L + 2\mu B)^2}{\lambda m^2}\left(t - 1 + \frac{(t-1)^2}{m-t+1}\right).$$

Plugging this back into Eq. (11), we get

$$\mathbb{E}[F(\mathbf{w}_t)] \leq \left(\frac{1}{2\eta_t} - \frac{\lambda}{4}\right)\mathbb{E}[\|\mathbf{w}_t\|^2] - \frac{1}{2\eta_t}\cdot\mathbb{E}[\|\mathbf{w}_{t+1}\|^2] + \frac{2(19L + 2\mu B)^2}{\lambda m^2}\left(t - 1 + \frac{(t-1)^2}{m-t+1}\right) + \frac{\eta_t}{2}G^2.$$

Averaging both sides over $t = 1, \ldots, T$, and using Jensen's inequality, we have

$$\mathbb{E}\left[\frac{1}{T}\sum_{t=1}^{T}F(\mathbf{w}_t)\right]$$

$$\leq \frac{1}{2T}\sum_{t=1}^{T}\mathbb{E}[\|\mathbf{w}_t\|^2]\left(\frac{1}{\eta_t} - \frac{1}{\eta_{t-1}} - \frac{\lambda}{2}\right) + \frac{2(19L+2\mu B)^2}{\lambda m^2 T}\sum_{t=1}^{T}\left(t-1+\frac{(t-1)^2}{m-t+1}\right) + \frac{G^2}{2T}\sum_{t=1}^{T}\eta_t,$$

where we use the convention that $1/\eta_0 = 0$. Since $T \leq m$, the second sum in the expression above equals

$$\sum_{t=0}^{T-1}\left(t+\frac{t^2}{m-t}\right) = \sum_{t=0}^{T-1}t + \sum_{t=0}^{T-1}\frac{t^2}{m-t} \leq \frac{T(T-1)}{2} + m^2\sum_{t=0}^{T-1}\frac{1}{m-t}$$

$$\leq \frac{m^2}{2} + m^2\left(\sum_{t=0}^{T-2}\frac{1}{m-t}+1\right) \leq \frac{3m^2}{2} + m^2\int_{t=0}^{T-1}\frac{1}{m-t}dt$$

$$= m^2\left(\frac{3}{2} + \log\left(\frac{m}{m-T+1}\right)\right).$$

Plugging this back in, we get

$$\mathbb{E}\left[\frac{1}{T}\sum_{t=1}^{T}F(\mathbf{w}_t)\right] \leq \frac{1}{2T}\sum_{t=1}^{T}\mathbb{E}[\|\mathbf{w}_t\|^2]\left(\frac{1}{\eta_t} - \frac{1}{\eta_{t-1}} - \frac{\lambda}{2}\right)$$

$$+ \frac{2(19L+2\mu B)^2\left(\frac{3}{2}+\log\left(\frac{m}{m-T+1}\right)\right)}{\lambda T} + \frac{G^2}{T}\sum_{t=1}^{T}\eta_t.$$

Now, choosing $\eta_t = 2/\lambda t$, and using the fact that $\sum_{t=1}^{T}\frac{1}{t} \leq \log(T)+1$, we get that

$$\mathbb{E}[F(\bar{\mathbf{w}}_T)] \leq \frac{2(19L+2\mu B)^2\left(\frac{3}{2}+\log\left(\frac{m}{m-T+1}\right)\right)}{\lambda T} + \frac{2G^2(\log(T)+1)}{\lambda T}.$$

The result follows by recalling that we assumed $F(\mathbf{w}^*) = F(\mathbf{0}) = 0$, and the observation that $T(m-T+1) \geq m$ for any $T \in \{1, 2, \ldots, m\}$, hence $\log(m/(m-T+1)) \leq \log(T)$.

### A.6 Proof of Thm. 4

The proof is based on propositions 1 and 2 presented below, which analyze the expectation of the update as well as its expected squared norm. The key technical challenge, required to get linear convergence, is to upper bound these quantities directly in terms of the suboptimality of the iterates $\mathbf{w}_t, \tilde{\mathbf{w}}_s$. To get Proposition 1, we state and prove a key lemma (Lemma 5 below), which bounds the without-replacement concentration behavior of certain normalized stochastic matrices. The proof of Thm. 4 itself is then a relatively straightforward calculation, relying on these results.

**Lemma 5.** *Let $\mathbf{x}_1, \ldots, \mathbf{x}_m$ be vectors in $\mathbb{R}^d$ of norm at most 1. Define $\bar{X} = \frac{1}{m}\sum_{i=1}^{m}\mathbf{x}_i\mathbf{x}_i^\top$, and*

$$M_i = (\bar{X}+\hat{\gamma}I)^{-1/2}\mathbf{x}_i\mathbf{x}_i^\top(\bar{X}+\hat{\gamma}I)^{-1/2}$$

*for some $\gamma \geq 0$, so that $\bar{X}+\hat{\gamma}I$ has minimal eigenvalue $\gamma \in (0,1)$. Finally, let $\sigma$ be a permutation on $\{1, \ldots, m\}$ drawn uniformly at random. Then for any $\alpha \geq 2$, the probability*

$$\Pr\left(\exists s \in \{1, \ldots, m\} : \left\|\frac{1}{s}\sum_{i=1}^{s}M_{\sigma(i)} - \frac{1}{m-s}\sum_{i=s+1}^{m}M_{\sigma(i)}\right\|\right.$$

$$\left. > \frac{\alpha}{\sqrt{\gamma}}\left(\frac{1}{\sqrt{s}}+\frac{1}{\sqrt{m-s}}\right) + \frac{\alpha}{\gamma}\left(\frac{1}{s}+\frac{1}{m-s}\right)\right)$$

*is at most $4dm\exp\left(-\alpha/2\right)$.*

*Proof.* The proof relies on a without-replacement version of Bernstein's inequality for matrices (Theorem 1 in [7]), which implies that for $d \times d$ Hermitican matrices $\hat{M}_i$ which satisfy

$$\frac{1}{m}\sum_{i=1}^m \hat{M}_i = 0 \ , \quad \max_i \left\|\hat{M}_i\right\| \le c \ , \quad \left\|\frac{1}{m}\sum_{i=1}^m \hat{M}_i^2\right\| \le v,$$

for some $v, c > 0$, it holds that

$$\Pr\left(\left\|\frac{1}{s}\sum_{i=1}^s \hat{M}_{\sigma(i)}\right\| > z\right) \ \le \ \begin{cases} 2d\exp\left(-\frac{sz^2}{4v}\right) & z \le 2v/c \\ 2d\exp\left(-\frac{sz}{2c}\right) & z > 2v/c \end{cases} \tag{12}$$

In particular, we will apply this on the matrices

$$\hat{M}_i \ = \ M_i - \frac{1}{m}\sum_{j=1}^m M_j \ = \ (\bar{X} + \gamma I)^{-1/2}\left(\mathbf{x}_i\mathbf{x}_i^\top - \bar{X}\right)(\bar{X} + \gamma I)^{-1/2}.$$

Clearly, $\frac{1}{m}\sum_{i=1}^m \hat{M}_i = 0$. We only need to find appropriate values for $v, c$.

First, by definition of $\hat{M}_i$, we have

$$\left\|\hat{M}_i\right\| \le \left\|(\bar{X} + \gamma I)^{-1/2}\right\| \left\|\mathbf{x}_i\mathbf{x}_i - \bar{X}\right\| \left\|(\bar{X} + \gamma I)^{-1/2}\right\|.$$

Since both $\bar{X}$ and $\mathbf{x}_i\mathbf{x}_i^\top$ are positive semidefinite and have spectral norm at most 1, the above is at most $\gamma^{-1/2} \cdot 1 \cdot \gamma^{-1/2} = \gamma^{-1}$. Therefore, we can take $c = 1/\gamma$.

We now turn to compute an appropriate value for $v$. For convenience, let $\mathbb{E}$ denote a uniform distribution over the index $i = 1, \ldots, m$, and note that $\mathbb{E}[\hat{M}_i] = 0$. Therefore, we have

$$\left\|\frac{1}{m}\sum_{i=1}^m \hat{M}_i^2\right\| = \left\|\mathbb{E}[\hat{M}_i^2]\right\| \ = \ \left\|\mathbb{E}[(M_i - \mathbb{E}[M_i])^2]\right\|$$

$$= \ \left\|\mathbb{E}[M_i^2] - \mathbb{E}^2[M_i]\right\| \ \le \ \max\{\left\|\mathbb{E}[M_i^2]\right\|, \left\|\mathbb{E}^2[M_i]\right\|\}, \tag{13}$$

where in the last step we used the fact that $M_i$ is positive semidefinite. Let us first upper bound the second term in the max, namely

$$\left\|\mathbb{E}^2[M_i]\right\| \ = \ \left\|\mathbb{E}[M_i]\cdot\mathbb{E}[M_i]\right\| \ \le \ \left\|\mathbb{E}[M_i]\right\|^2 \ = \ \left\|(\bar{X} + \gamma I)^{-1/2}\bar{X}(\bar{X} + \gamma I)^{-1/2}\right\|^2.$$

Since the expression above is invariant to rotating the positive semidefinite matrix $\bar{X}$, we can assume without loss of generality that $\bar{X} = \text{diag}(s_1, \ldots, s_d)$, in which case the above reduces to $\left(\max_i \frac{s_i}{s_i + \gamma}\right)^2 \le 1$. Turning to the first term in the max in Eq. (13), we have

$$\left\|\mathbb{E}[M_i^2]\right\| = \left\|\frac{1}{m}\sum_{i=1}^m (\bar{X} + \gamma I)^{-1/2}\mathbf{x}_i\mathbf{x}_i^\top(\bar{X} + \gamma I)^{-1}\mathbf{x}_i\mathbf{x}_i^\top(\bar{X} + \gamma I)^{-1/2}\right\|$$

$$= \left\|\frac{1}{m}\sum_{i=1}^m \left(\mathbf{x}_i^\top(\bar{X} + \gamma I)^{-1}\mathbf{x}_i\right)(\bar{X} + \gamma I)^{-1/2}\mathbf{x}_i\mathbf{x}_i^\top(\bar{X} + \gamma I)^{-1/2}\right\|$$

$$\overset{(1)}{\le} \left\|\frac{1}{m}\sum_{i=1}^m \left\|(\bar{X} + \gamma I)^{-1}\right\|(\bar{X} + \gamma I)^{-1/2}\mathbf{x}_i\mathbf{x}_i^\top(\bar{X} + \gamma I)^{-1/2}\right\|$$

$$= \left\|(\bar{X} + \gamma I)^{-1}\right\|\left\|(\bar{X} + \gamma I)^{-1/2}\bar{X}(\bar{X} + \gamma I)^{-1/2}\right\| \tag{14}$$

where in (1) we used the facts that $\|\mathbf{x}_i\| \le 1$ and each term $(\bar{X} + \gamma I)^{-1/2}\mathbf{x}_i\mathbf{x}_i^\top(\bar{X} + \gamma I)^{-1/2}$ is positive semidefinite. As before, the expression above is invariant to rotating the positive semidefinite matrix $\bar{X}$, so we can assume without loss of generality that $\bar{X} = \text{diag}(s_1, \ldots, s_d)$, in which case the above reduces to

$$\left(\max_i \frac{1}{s_i + \gamma}\right)\left(\max_i \frac{s_i}{s_i + \gamma}\right) \ \le \ \frac{1}{\gamma}\cdot 1 \ = \ \frac{1}{\gamma}.$$

Plugging these observations back into Eq. (13), we get that

$$\left\| \frac{1}{m} \sum_{i=1}^{m} \hat{M}_i^2 \right\| \leq \max\left\{ 1, \frac{1}{\gamma} \right\} = \frac{1}{\gamma}.$$

Therefore, Eq. (12) applies with $v = c = 1/\gamma$,, so we get that

$$\Pr\left( \left\| \frac{1}{s} \sum_{i=1}^{s} \hat{M}_{\sigma(i)} \right\| > z \right) \leq \begin{cases} 2d\exp\left( -\frac{\gamma s z^2}{4} \right) & z \leq 2 \\ 2d\exp\left( -\frac{\gamma s z}{2} \right) & z > 2 \end{cases}.$$

Substituting $z = \alpha\left( \frac{1}{\sqrt{\gamma s}} + \frac{1}{\gamma s} \right)$, we get that $\Pr\left( \left\| \frac{1}{s} \sum_{i=1}^{s} \hat{M}_{\sigma(i)} \right\| > \alpha\left( \frac{1}{\sqrt{\gamma s}} + \frac{1}{\gamma s} \right) \right)$ can be upper bounded by

$$2d\exp\left( -\frac{1}{4}\gamma s \alpha^2 \left( \frac{1}{\sqrt{\gamma s}} + \frac{1}{\gamma s} \right)^2 \right) \leq 2d\exp\left( -\frac{\alpha^2}{4} \right)$$

in the first case, and

$$2d\exp\left( -\frac{1}{2}\gamma s \alpha \left( \frac{1}{\sqrt{\gamma s}} + \frac{1}{\gamma s} \right) \right) \leq 2d\exp\left( -\frac{\alpha}{2} \right)$$

in the second case. Assuming $\alpha \geq 2$, both expressions can be upper bounded by $2d\exp(-\alpha/2)$, so we get that

$$\Pr\left( \left\| \frac{1}{s} \sum_{i=1}^{s} \hat{M}_{\sigma(i)} \right\| > \alpha\left( \frac{1}{\sqrt{\gamma s}} + \frac{1}{\gamma s} \right) \right) \leq 2d\exp\left( -\frac{\alpha}{2} \right)$$

for any $\alpha \geq 2$. Recalling the definition of $\hat{M}_i$, we get

$$\Pr\left( \left\| \frac{1}{s} \sum_{i=1}^{s} M_{\sigma(i)} - \frac{1}{m} \sum_{i=1}^{m} M_i \right\| > \alpha\left( \frac{1}{\sqrt{\gamma s}} + \frac{1}{\gamma s} \right) \right) \leq 2d\exp\left( -\frac{\alpha}{2} \right). \qquad (15)$$

Since the permutation is random, the exact same line of argument also works if we consider the last $m - s$ matrices rather than the first $s$ matrices, that is

$$\Pr\left( \left\| \frac{1}{m-s} \sum_{i=s+1}^{m} M_{\sigma(i)} - \frac{1}{m} \sum_{i=1}^{m} M_i \right\| > \alpha\left( \frac{1}{\sqrt{\gamma(m-s)}} + \frac{1}{\gamma(m-s)} \right) \right) \leq 2d\exp\left( -\frac{\alpha}{2} \right). \qquad (16)$$

Now, notice that for any matrices $A, B, C$ and scalars $a, b$, it holds that

$$\Pr(\|A - B\| > a + b) \leq \Pr(\|A - C\| > a) + \Pr(\|B - C\| > b)$$

(as the event $\|A - B\| > a + b$ implies $\|A - C\| + \|B - C\| > a + b$). Using this observation and Eq. (15), Eq. (16), we have

$$\Pr\left( \left\| \frac{1}{s} \sum_{i=1}^{s} M_{\sigma(i)} - \frac{1}{m-s} \sum_{i=s+1}^{m} M_{\sigma(i)} \right\| > \frac{\alpha}{\sqrt{\gamma}} \left( \frac{1}{\sqrt{s}} + \frac{1}{\sqrt{m-s}} \right) + \frac{\alpha}{\gamma}\left( \frac{1}{s} + \frac{1}{m-s} \right) \right)$$

$$\leq \Pr\left( \left\| \frac{1}{s} \sum_{i=1}^{s} M_{\sigma(i)} - \frac{1}{m} \sum_{i=1}^{m} M_i \right\| > \alpha\left( \frac{1}{\sqrt{\gamma s}} + \frac{1}{\gamma s} \right) \right)$$

$$+ \Pr\left( \left\| \frac{1}{m-s} \sum_{i=s+1}^{m} M_{\sigma(i)} - \frac{1}{m} \sum_{i=1}^{m} M_i \right\| > \alpha\left( \frac{1}{\sqrt{\gamma(m-s)}} + \frac{1}{\gamma(m-s)} \right) \right)$$

$$\leq 4d\exp\left( -\frac{\alpha}{2} \right).$$

The statement in the lemma now follows from a union bound argument over all possible $s = 1, 2, \ldots, m$. $\qquad \square$

**Proposition 1.** *Suppose each $f_i(\cdot)$ is of the form in Eq. (2), where $\mathbf{x}_i, \mathbf{w}$ are in $\mathbb{R}^d$, and $F(\cdot)$ is $\lambda$-strongly convex with $\lambda \in [1/m, 1]$. Define*

$$\mathbf{v}_i(t, s) = \nabla f_i(\mathbf{w}_t) - \nabla f_i(\tilde{\mathbf{w}}_s) + \nabla F(\tilde{\mathbf{w}}_s).$$

*Then for any $t \leq m/2$ and any $\epsilon \in (0, 1)$,*

$$\mathbb{E}\left[\langle \mathbf{v}_{\sigma(t)}(t, s), \mathbf{w}_t - \mathbf{w}^* \rangle - \frac{1}{m}\sum_{i=1}^m \langle \mathbf{v}_i(t, s), \mathbf{w}_t - \mathbf{w}^* \rangle\right]$$

$$\leq \frac{\epsilon}{2} + \frac{18}{\sqrt{\lambda m}} \log\left(\frac{64 dm B^2}{\lambda \epsilon}\right) \cdot \mathbb{E}\left[F(\mathbf{w}_t) + F(\tilde{\mathbf{w}}_s) - 2F(\mathbf{w}^*)\right],$$

*where $d$ is the dimension.*

*Proof.* Define

$$\mathbf{u}_i(t, s) = \langle \mathbf{v}_i(t, s), \mathbf{w}_t - \mathbf{w}^* \rangle = \langle \nabla f_i(\mathbf{w}_t) - \nabla f_i(\tilde{\mathbf{w}}_s) + \nabla F(\tilde{\mathbf{w}}_s), \, \mathbf{w}_t - \mathbf{w}^* \rangle,$$

in which case the expectation in the proposition statement equals

$$\mathbb{E}\left[\mathbf{u}_{\sigma(t)}(t, s) - \frac{1}{m}\sum_{i=1}^m \mathbf{u}_i(t, s)\right].$$

Notice that $\mathbf{u}_i(t, s)$ for all $i$ is independent of $\sigma(t), \ldots, \sigma(m)$ conditioned on $\sigma(1), \ldots, \sigma(t-1)$ (which determine $\mathbf{w}_t$ and $\tilde{\mathbf{w}}_s$). Therefore, we can apply Lemma 1, and get that the above equals

$$\frac{t-1}{m} \cdot \mathbb{E}\left[\mathbf{u}_{t:m}(t, s) - \mathbf{u}_{1:t-1}(t, s)\right]. \tag{17}$$

Recalling the definition of $\mathbf{u}_i(t, s)$, and noting that the fixed $\langle \nabla F(\tilde{\mathbf{w}}_s), \mathbf{w}_t - \mathbf{w}^* \rangle$ terms get cancelled out in the difference above, we get that Eq. (17) equals

$$\frac{t-1}{m} \cdot \mathbb{E}\left[\check{\mathbf{u}}_{t:m}(t, s) - \check{\mathbf{u}}_{1:t-1}(t, s)\right]. \tag{18}$$

where

$$\check{\mathbf{u}}_i(t, s) = \langle \nabla f_i(\mathbf{w}_t) - \nabla f_i(\tilde{\mathbf{w}}_s), \, \mathbf{w}_t - \mathbf{w}^* \rangle$$

$$= \langle \mathbf{w}_t - \tilde{\mathbf{w}}_s, \mathbf{x}_i \rangle \cdot \langle \mathbf{x}_i, \, \mathbf{w}_t - \mathbf{w}^* \rangle + \hat{\lambda} \langle \mathbf{w}_t - \tilde{\mathbf{w}}_s, \mathbf{w}_t - \mathbf{w}^* \rangle.$$

Again, the fixed $\hat{\lambda} \langle \mathbf{w}_t - \tilde{\mathbf{w}}_s, \mathbf{w}_t - \mathbf{w}^* \rangle$ terms get cancelled out in Eq. (18), so we can rewrite Eq. (18) as

$$\frac{t-1}{m} \cdot \mathbb{E}\left[\breve{\mathbf{u}}_{t:m}(t, s) - \breve{\mathbf{u}}_{1:t-1}(t, s)\right] \tag{19}$$

where

$$\breve{\mathbf{u}}_i(t, s) = \langle \mathbf{w}_t - \tilde{\mathbf{w}}_s, \mathbf{x}_i \rangle \cdot \langle \mathbf{x}_i, \, \mathbf{w}_t - \mathbf{w}^* \rangle$$

$$= \langle \mathbf{w}_t - \mathbf{w}^*, \mathbf{x}_i \rangle \cdot \langle \mathbf{x}_i, \mathbf{w}_t - \mathbf{w}^* \rangle + \langle \mathbf{w}^* - \tilde{\mathbf{w}}_s, \mathbf{x}_i \rangle \cdot \langle \mathbf{x}_i, \mathbf{w}_t - \mathbf{w}^* \rangle$$

$$= (\mathbf{w}_t - \mathbf{w}^*)^\top \mathbf{x}_i \mathbf{x}_i^\top (\mathbf{w}_t - \mathbf{w}^*) + (\mathbf{w}^* - \tilde{\mathbf{w}}_s)^\top \mathbf{x}_i \mathbf{x}_i^\top (\mathbf{w}_t - \mathbf{w}^*).$$

Therefore, we can rewrite Eq. (19) as

$$\frac{t-1}{m} \cdot \mathbb{E}\left[(\mathbf{w}_t - \mathbf{w}^*)^\top \left(\frac{1}{m-t+1}\sum_{i=t}^m \mathbf{x}_{\sigma(i)}\mathbf{x}_{\sigma(i)}^\top - \frac{1}{t-1}\sum_{i=1}^{t-1} \mathbf{x}_{\sigma(i)}\mathbf{x}_{\sigma(i)}^\top\right)(\mathbf{w}_t - \mathbf{w}^*)\right]$$

$$+ \frac{t-1}{m} \cdot \mathbb{E}\left[(\mathbf{w}^* - \tilde{\mathbf{w}}_s)^\top \left(\frac{1}{m-t+1}\sum_{i=t}^m \mathbf{x}_{\sigma(i)}\mathbf{x}_{\sigma(i)}^\top - \frac{1}{t-1}\sum_{i=1}^{t-1} \mathbf{x}_{\sigma(i)}\mathbf{x}_{\sigma(i)}^\top\right)(\mathbf{w}_t - \mathbf{w}^*)\right]$$

$$\tag{20}$$

To continue, note that for any symmetric square matrix $M$, positive semidefinite matrix $A$, and vectors $\mathbf{w}_1, \mathbf{w}_2$, we have

$$
|\mathbf{w}_1^\top M \mathbf{w}_2| = \left(\mathbf{w}_1^\top A \mathbf{w}_1 + \mathbf{w}_2^\top A \mathbf{w}_2\right) \left| \frac{\mathbf{w}_1^\top M \mathbf{w}_2}{\mathbf{w}_1^\top A \mathbf{w}_1 + \mathbf{w}_2^\top A \mathbf{w}_2} \right|
$$

$$
\leq \left(\mathbf{w}_1^\top A \mathbf{w}_1 + \mathbf{w}_2^\top A \mathbf{w}_2\right) \sup_{\mathbf{w}_1, \mathbf{w}_2} \left| \frac{\mathbf{w}_1^\top M \mathbf{w}_2}{\mathbf{w}_1^\top A \mathbf{w}_1 + \mathbf{w}_2^\top A \mathbf{w}_2} \right|
$$

$$
\overset{(1)}{\leq} \left(\mathbf{w}_1^\top A \mathbf{w}_1 + \mathbf{w}_2^\top A \mathbf{w}_2\right) \sup_{\mathbf{w}_1, \mathbf{w}_2} \left| \frac{\mathbf{w}_1^\top A^{-1/2} M A^{-1/2} \mathbf{w}_2}{\|\mathbf{w}_1\|^2 + \|\mathbf{w}_2\|^2} \right|
$$

$$
\overset{(2)}{\leq} \left(\mathbf{w}_1^\top A \mathbf{w}_1 + \mathbf{w}_2^\top A \mathbf{w}_2\right) \sup_{\mathbf{w}_1, \mathbf{w}_2} \left| \frac{\mathbf{w}_1^\top A^{-1/2} M A^{-1/2} \mathbf{w}_2}{2\|\mathbf{w}_1\| \|\mathbf{w}_2\|} \right|
$$

$$
\overset{(3)}{\leq} \frac{1}{2} \left(\mathbf{w}_1^\top A \mathbf{w}_1 + \mathbf{w}_2^\top A \mathbf{w}_2\right) \left\| A^{-1/2} M A^{-1/2} \right\|,
$$

where (1) is by substituting $A^{-1/2} \mathbf{w}_1, A^{-1/2} \mathbf{w}_2$ in lieu of $\mathbf{w}_1, \mathbf{w}_2$ in the supremum, (2) is by the identity $a^2 + b^2 \geq 2ab$, and (3) is by the fact that for any square matrix $X$, $|\mathbf{w}_1^\top X \mathbf{w}_2| \leq \|\mathbf{w}_1\| \|X\| \|\mathbf{w}_2\|$. Applying this inequality with

$$
M = \frac{1}{m-t+1} \sum_{i=t}^{m} \mathbf{x}_{\sigma(i)} \mathbf{x}_{\sigma(i)}^\top - \frac{1}{t-1} \sum_{i=1}^{t-1} \mathbf{x}_{\sigma(i)} \mathbf{x}_{\sigma(i)}^\top \ , \quad A = \frac{1}{m} \sum_{i=1}^{m} \mathbf{x}_i \mathbf{x}_i^\top + \frac{\hat{\lambda}}{2} I,
$$

$\mathbf{w}_1$ being either $\mathbf{w}^* - \tilde{\mathbf{w}}_s$ or $\mathbf{w}_t - \mathbf{w}^*$, and $\mathbf{w}_2 = \mathbf{w}_t - \mathbf{w}^*$, we can (somewhat loosely) upper bound Eq. (20) by

$$
\frac{3(t-1)}{2m} \cdot \mathbb{E}\left[ \left((\mathbf{w}_t - \mathbf{w}^*)^\top A (\mathbf{w}_t - \mathbf{w}^*) + (\tilde{\mathbf{w}}_s - \mathbf{w}^*)^\top A (\tilde{\mathbf{w}}_s - \mathbf{w}^*)\right) \left\| A^{-1/2} M A^{-1/2} \right\| \right].
$$

Recalling that the objective function $F(\cdot)$ is actually of the form $F(\mathbf{w}) = \mathbf{w}^\top A \mathbf{w} + \mathbf{b}^\top \mathbf{w} + c$ for the positive definite matrix $A$ as above, and some vector $\mathbf{b}$ and scalar $c$, it is easily verified that $\mathbf{w}^* = -\frac{1}{2} A^{-1} \mathbf{b}$, and moreover, that

$$
(\mathbf{w} - \mathbf{w}^*)^\top A (\mathbf{w} - \mathbf{w}^*) = F(\mathbf{w}) - F(\mathbf{w}^*)
$$

for any $\mathbf{w}$. Therefore, we can rewrite the above as

$$
\frac{3(t-1)}{2m} \cdot \mathbb{E}\left[ (F(\mathbf{w}_t) + F(\tilde{\mathbf{w}}_s) - 2F(\mathbf{w}^*)) \left\| A^{-1/2} M A^{-1/2} \right\| \right]
$$

$$
= \frac{3(t-1)}{2m} \cdot \mathbb{E}\left[ (F(\mathbf{w}_t) + F(\tilde{\mathbf{w}}_s) - 2F(\mathbf{w}^*)) \left\| A^{-1/2} \left( \frac{\sum_{i=t}^{m} \mathbf{x}_{\sigma(i)} \mathbf{x}_{\sigma(i)}^\top}{m-t+1} - \frac{\sum_{i=1}^{t-1} \mathbf{x}_{\sigma(i)} \mathbf{x}_{\sigma(i)}^\top}{t-1} \right) A^{-1/2} \right\| \right].
$$

$$\tag{21}$$

We now wish to use Lemma 5, which upper bounds the norm in the expression above with high probability. However, since the norm appears inside an expectation and is multiplied by another term, we need to proceed a bit more carefully. To that end, let $N$ denote the norm in the expression above, and let $D$ denote the expression $F(\mathbf{w}_t) + F(\tilde{\mathbf{w}}_s) - 2F(\mathbf{w}^*)$. We collect the following observations:

- The Hessian of the objective function $F(\cdot)$ is $\frac{1}{m} \sum_{i=1}^{m} \mathbf{x}_i \mathbf{x}_i^\top + \hat{\lambda} I$, whose minimal eigenvalue is at least $\lambda$ (since $F(\cdot)$ is assumed to be $\lambda$-strongly convex). Therefore, the minimal eigenvalue of $A$ as defined above is at least $\lambda/2$. Applying Lemma 5, $\Pr(N > \alpha \cdot q(t)) \leq 2dm \exp(-\alpha/2)$ for any $\alpha \geq 2$, where

$$
q(t) = \sqrt{\frac{2}{\lambda}} \left( \frac{1}{\sqrt{t-1}} + \frac{1}{\sqrt{m-t+1}} \right) + \frac{2}{\lambda} \left( \frac{1}{t-1} + \frac{1}{m-t+1} \right).
$$

  for any $t > 1$, and $q(1) = 0$.

- By assumption, $\|\mathbf{w}_t\|, \|\tilde{\mathbf{w}}_s\|$ and $\|\mathbf{w}^*\|$ are all at most $B$. Moreover, since the objective function $F(\cdot)$ is $1 + \hat{\lambda} \leq 2$ smooth, $F(\mathbf{w}_t) - F(\mathbf{w}^*) \leq \|\mathbf{w}_t - \mathbf{w}^*\|^2 \leq q4B^2$ and $F(\tilde{\mathbf{w}}_s) - F(\mathbf{w}^*) \leq \|\tilde{\mathbf{w}}_s - \mathbf{w}^*\|^2 \leq 4B^2$. As a result, $D$ as defined above is in $[0, 8B^2]$.

- Since each $\mathbf{x}_i\mathbf{x}_i^\top$ has spectral norm at most 1, $N$ is at most $\left\|A^{-1/2}\right\|^2 \leq \frac{2}{\lambda}$.

Combining these observations, we have the following:

$$\mathbb{E}[DN] = \Pr(N > \alpha q(t)) \cdot \mathbb{E}[DN|N > \alpha q(t)] + \Pr(N \leq \alpha q(t)) \cdot \mathbb{E}[DN|N \leq \alpha q(t)]$$

$$\leq 2dm \exp\left(-\frac{\alpha}{2}\right)\frac{16B^2}{\lambda} + \alpha q(t) \cdot \Pr(N \leq \alpha q(t)) \cdot \mathbb{E}[D|N \leq \alpha q(t)]$$

$$\leq \frac{32dmB^2}{\lambda} \exp\left(-\frac{\alpha}{2}\right) + \alpha q(t) \cdot \mathbb{E}[D]$$

for any $\alpha \geq 2$. In particular, picking $\alpha = 2\log(64dmB^2/\lambda\epsilon)$ (where recall that $\epsilon \in (0,1)$ is an arbitrary parameter), we get

$$\mathbb{E}[DN] \leq \frac{\epsilon}{2} + 2\log\left(\frac{64dmB^2}{\lambda\epsilon}\right) q(t) \cdot \mathbb{E}[D].$$

Plugging in the definition of $D$, we get the following upper bound on Eq. (21):

$$\frac{3(t-1)}{2m}\left(\frac{\epsilon}{2} + 2\log\left(\frac{64dmB^2}{\lambda\epsilon}\right)q(t)\cdot \mathbb{E}\left[F(\mathbf{w}_t) + F(\tilde{\mathbf{w}}_s) - 2F(\mathbf{w}^*)\right]\right). \tag{22}$$

Recalling the definition of $q(t)$ and the assumption $t \leq m/2$ (and using the convention $0/\sqrt{0} = 0$), we have

$$\frac{3(t-1)}{2m} \cdot q(t) = \frac{3(t-1)}{2m}\left(\sqrt{\frac{2}{\lambda}}\left(\frac{1}{\sqrt{t-1}} + \frac{1}{\sqrt{m-t+1}}\right) + \frac{3}{\lambda}\left(\frac{1}{t-1} + \frac{1}{m-t+1}\right)\right)$$

$$= \frac{3}{\sqrt{2\lambda}}\left(\frac{\sqrt{t-1}}{m} + \frac{t-1}{m\sqrt{m-t+1}}\right) + \frac{3}{\lambda}\left(\frac{1}{m} + \frac{t-1}{m(m-t+1)}\right)$$

$$\leq \frac{3}{\sqrt{2\lambda}}\left(\frac{\sqrt{m/2}}{m} + \frac{m/2}{m\sqrt{m/2}}\right) + \frac{3}{\lambda}\left(\frac{1}{m} + \frac{m/2}{m(m/2)}\right)$$

$$= \frac{3}{\sqrt{2\lambda}}\left(\frac{1}{\sqrt{2m}} + \frac{1}{\sqrt{2m}}\right) + \frac{3}{\lambda}\left(\frac{1}{m} + \frac{1}{m}\right)$$

$$= \frac{3}{\sqrt{\lambda m}} + \frac{6}{\lambda m} = \frac{3}{\sqrt{\lambda m}}\left(1 + \frac{2}{\sqrt{\lambda m}}\right),$$

which by the assumption $\lambda \geq 1/m$ (hence $\lambda m \geq 1$), is at most $9/\sqrt{\lambda m}$. Substituting this back into Eq. (22) and loosely upper bounding, we get the upper bound

$$\frac{\epsilon}{2} + \frac{18}{\sqrt{\lambda m}}\log\left(\frac{64dmB^2}{\lambda\epsilon}\right) \cdot \mathbb{E}\left[F(\mathbf{w}_t) + F(\tilde{\mathbf{w}}_s) - 2F(\mathbf{w}^*)\right],$$

as required. $\square$

**Proposition 2.** *Let*

$$\mathbf{v}_i(t,s) = \nabla f_i(\mathbf{w}_t) - \nabla f_i(\tilde{\mathbf{w}}_s) + \nabla F(\tilde{\mathbf{w}}_s).$$

*and suppose each $f_i(\cdot)$ is $\mu$-smooth. Then for any $t \leq m/2$,*

$$\mathbb{E}[\left\|\mathbf{v}_{\sigma(t)}(t,s)\right\|^2] \leq 6\mu(F(\mathbf{w}_t) + F(\tilde{\mathbf{w}}_s) - 2F(\mathbf{w}^*))$$

*Proof.* Since $\nabla F(\mathbf{w}^*) = \mathbf{0}$, we can rewrite $\mathbf{v}_{\sigma(t)}(t,s)$ as

$$g_{\sigma(t)}(\mathbf{w}_t) - g_{\sigma(t)}(\tilde{\mathbf{w}}_s) + (\nabla F(\tilde{\mathbf{w}}_s) - \nabla F(\mathbf{w}^*)),$$

where

$$g_{\sigma(t)}(\mathbf{w}) = \nabla f_{\sigma(t)}(\mathbf{w}) - \nabla f_{\sigma(t)}(\mathbf{w}^*).$$

Using the fact that $(a + b + c)^2 \leq 3(a^2 + b^2 + c^2)$ for any $a, b, c$, we have

$$\frac{1}{3}\mathbb{E}\left[\left\|\mathbf{v}_{\sigma(t)}(t,s)\right\|^2\right] \tag{23}$$

$$\leq \mathbb{E}\left[\left\|g_{\sigma(t)}(\mathbf{w}_t)\right\|^2\right] + \mathbb{E}\left[\left\|g_{\sigma(t)}(\tilde{\mathbf{w}}_s)\right\|^2\right] + \mathbb{E}\left[\left\|\nabla F(\tilde{\mathbf{w}}_s) - \nabla F(\mathbf{w}^*)\right\|^2\right]$$

$$= \mathbb{E}\left[\frac{1}{m}\sum_{i=1}^m \|g_i(\mathbf{w}_t)\|^2\right] + \mathbb{E}\left[\frac{1}{m}\sum_{i=1}^m \|g_i(\tilde{\mathbf{w}}_s)\|^2\right] + \mathbb{E}\left[\left\|\nabla F(\tilde{\mathbf{w}}_s) - \nabla F(\mathbf{w}^*)\right\|^2\right]$$

$$+ \mathbb{E}\left[\left\|g_{\sigma(t)}(\mathbf{w}_t)\right\|^2 - \frac{1}{m}\sum_{i=1}^m \|g_i(\mathbf{w}_t)\|^2\right] + \mathbb{E}\left[\left\|g_{\sigma(t)}(\tilde{\mathbf{w}}_s)\right\|^2 - \frac{1}{m}\sum_{i=1}^m \|g_i(\tilde{\mathbf{w}}_s)\|^2\right] \tag{24}$$

We now rely on a simple technical result proven as part of the standard SVRG analysis (see equation (8) in [11]), which states that if $P(\mathbf{w}) = \frac{1}{n}\sum_{i=1}^n \psi_i(\mathbf{w})$, where each $\psi_i$ is convex and $\mu$-smooth, and $P$ is minimized at $\mathbf{w}^*$, then for all $\mathbf{w}$.

$$\frac{1}{n}\sum_{i=1}^n \|\nabla\psi_i(\mathbf{w}) - \nabla\psi_i(\mathbf{w}^*)\|^2 \leq 2\mu\left(P(\mathbf{w}) - P(\mathbf{w}^*)\right) \tag{25}$$

Applying this inequality on each of the first 3 terms in Eq. (24) (i.e. taking either $\psi_i(\cdot) = f_i(\cdot)$ and $n = m$, or $\psi(\cdot) = F(\cdot)$ and $n = 1$), we get the upper bound

$$2\mu(F(\mathbf{w}_t) - F(\mathbf{w}^*)) + 2\mu(F(\tilde{\mathbf{w}}_s) - F(\mathbf{w}^*)) + 2\mu(F(\tilde{\mathbf{w}}_s) - F(\mathbf{w}^*))$$

$$+ \mathbb{E}\left[\left\|g_{\sigma(t)}(\mathbf{w}_t)\right\|^2 - \frac{1}{m}\sum_{i=1}^m \|g_i(\mathbf{w}_t)\|^2\right] + \mathbb{E}\left[\left\|g_{\sigma(t)}(\tilde{\mathbf{w}}_s)\right\|^2 - \frac{1}{m}\sum_{i=1}^m \|g_i(\tilde{\mathbf{w}}_s)\|^2\right]$$

$$= 2\mu\left(F(\mathbf{w}_t) - F(\mathbf{w}^*)\right) + 4\mu\left(F(\tilde{\mathbf{w}}_s) - F(\mathbf{w}^*)\right)$$

$$+ \mathbb{E}\left[\left\|g_{\sigma(t)}(\mathbf{w}_t)\right\|^2 - \frac{1}{m}\sum_{i=1}^m \|g_i(\mathbf{w}_t)\|^2\right] + \mathbb{E}\left[\left\|g_{\sigma(t)}(\tilde{\mathbf{w}}_s)\right\|^2 - \frac{1}{m}\sum_{i=1}^m \|g_i(\tilde{\mathbf{w}}_s)\|^2\right].$$

Loosely upper bounding this and applying Lemma 1, we get the upper bound

$$4\mu\left(F(\mathbf{w}_t) + F(\tilde{\mathbf{w}}_s) - 2F(\mathbf{w}^*)\right) + \frac{t-1}{m}\cdot\mathbb{E}\left[\frac{\sum_{i=t}^m \left\|g_{\sigma(i)}(\mathbf{w}_t)\right\|}{m-t+1} - \frac{\sum_{i=1}^{t-1}\left\|g_{\sigma(i)}(\mathbf{w}_t)\right\|^2}{t-1}\right]$$

$$+ \frac{t-1}{m}\cdot\mathbb{E}\left[\frac{\sum_{i=t}^m \left\|g_{\sigma(i)}(\tilde{\mathbf{w}}_s)\right\|}{m-t+1} - \frac{\sum_{i=1}^{t-1}\left\|g_{\sigma(i)}(\tilde{\mathbf{w}}_s)\right\|^2}{t-1}\right]$$

$$\leq 4\mu\left(F(\mathbf{w}_t) + F(\tilde{\mathbf{w}}_s) - 2F(\mathbf{w}^*)\right) + \frac{t-1}{m}\cdot\mathbb{E}\left[\frac{\sum_{i=t}^m \left\|g_{\sigma(i)}(\mathbf{w}_t)\right\|^2}{m-t+1} + \frac{\sum_{i=t}^m \left\|g_{\sigma(t)}(\tilde{\mathbf{w}}_s)\right\|^2}{m-t+1}\right]$$

$$\leq 4\mu\left(F(\mathbf{w}_t) + F(\tilde{\mathbf{w}}_s) - 2F(\mathbf{w}^*)\right) + \frac{t-1}{m}\cdot\mathbb{E}\left[\frac{\sum_{i=1}^m \left\|g_{\sigma(i)}(\mathbf{w}_t)\right\|^2}{m-t+1} + \frac{\sum_{i=1}^m \left\|g_{\sigma(i)}(\tilde{\mathbf{w}}_s)\right\|^2}{m-t+1}\right]$$

$$= 4\mu\left(F(\mathbf{w}_t) + F(\tilde{\mathbf{w}}_s) - 2F(\mathbf{w}^*)\right) + \frac{t-1}{m-t+1}\cdot\mathbb{E}\left[\frac{\sum_{i=1}^m \left\|g_i(\mathbf{w}_t)\right\|^2}{m} + \frac{\sum_{i=1}^m \left\|g_i(\tilde{\mathbf{w}}_s)\right\|^2}{m}\right].$$

Since we assume $t \leq m/2$, we have $\frac{t-1}{m-t+1} \leq \frac{m/2}{m/2} = 1$. Plugging this in, and applying Eq. (25) on the $\frac{1}{m}\sum_{i=1}^m \left\|g_{\sigma(i)}(\mathbf{w}_t)\right\|^2$ and $\frac{1}{m}\sum_{i=1}^m \left\|g_{\sigma(i)}(\tilde{\mathbf{w}}_s)\right\|^2$ terms, this is at most

$$4\mu\left(F(\mathbf{w}_t) + F(\tilde{\mathbf{w}}_s) - 2F(\mathbf{w}^*)\right) + 1 \cdot \left(2\mu(F(\mathbf{w}_t) - F(\mathbf{w}^*)) + 2\mu(F(\tilde{\mathbf{w}}_s) - F(\mathbf{w}^*))\right)$$

$$= 6\mu(F(\mathbf{w}_t) + F(\tilde{\mathbf{w}}_s) - 2F(\mathbf{w}^*))$$

as required. $\qquad\square$

*Proof of Thm. 4.* Consider some specific epoch $s$ and iteration $t$. We have

$$\mathbf{w}_{t+1} = \mathbf{w}_t - \mathbf{v}_{\sigma(t)}(t,s),$$

where
$$\mathbf{v}_i(t, s) = \nabla f_i(\mathbf{w}_t) - \nabla f_i(\tilde{\mathbf{w}}_s) + \nabla F(\tilde{\mathbf{w}}_s).$$

Therefore,

$$
\begin{aligned}
\mathbb{E}[\|\mathbf{w}_{t+1} - \mathbf{w}^*\|^2] &= \mathbb{E}[\|\mathbf{w}_t - \eta \mathbf{v}_{\sigma(t)}(t, s)\|^2] \\
&= \mathbb{E}[\|\mathbf{w}_t - \mathbf{w}^*\|^2] - 2\eta \cdot \mathbb{E}[\langle \mathbf{v}_{\sigma(t)}(t, s), \mathbf{w}_t - \mathbf{w}^* \rangle] + \eta^2 \mathbb{E}[\|\mathbf{v}_{\sigma(t)}(t, s)\|^2].
\end{aligned}
$$
(26)

Applying Proposition 1 and Proposition 2 (assuming that $t \leq m/2$, which we will verify later, and noting that $\lambda \geq 1/m$ by the assumptions on $\eta, T$ and $m$, and that each $f_i(\cdot)$ is $1 + \hat{\lambda} \leq 2$-smooth), Eq. (26) is at most

$$
\begin{aligned}
& \mathbb{E}[\|\mathbf{w}_t - \mathbf{w}^*\|^2] - 2\eta \cdot \mathbb{E}\left[\frac{1}{m}\sum_{i=1}^{m} \langle \mathbf{v}_i(t, s), \mathbf{w}_t - \mathbf{w}^* \rangle\right] + 12\eta^2(F(\mathbf{w}_t) + F(\tilde{\mathbf{w}}_s) - 2F(\mathbf{w}^*)) \\
& \quad + 2\eta\left(\frac{\epsilon}{2} + \frac{18}{\sqrt{\lambda m}}\log\left(\frac{64dmB^2}{\lambda\epsilon}\right) \cdot \mathbb{E}[F(\mathbf{w}_t) + F(\tilde{\mathbf{w}}_s) - 2F(\mathbf{w}^*)]\right) \\
&= \mathbb{E}[\|\mathbf{w}_t - \mathbf{w}^*\|^2] - 2\eta \cdot \mathbb{E}[\langle \nabla F(\mathbf{w}_t), \mathbf{w}_t - \mathbf{w}^* \rangle] \\
& \quad + \eta\epsilon + 2\eta\left(6\eta + \frac{18}{\sqrt{\lambda m}}\log\left(\frac{64dmB^2}{\lambda\epsilon}\right)\right) \cdot \mathbb{E}[F(\mathbf{w}_t) + F(\tilde{\mathbf{w}}_s) - 2F(\mathbf{w}^*)].
\end{aligned}
$$

Since $F(\cdot)$ is convex, $\langle \nabla F(\mathbf{w}_t), \mathbf{w}_t - \mathbf{w}^* \rangle \geq F(\mathbf{w}_t) - F(\mathbf{w}^*)$, so we can upper bound the above by

$$
\begin{aligned}
& \mathbb{E}[\|\mathbf{w}_t - \mathbf{w}^*\|^2] + \eta\epsilon + 2\eta\left(6\eta + \frac{18}{\sqrt{\lambda m}}\log\left(\frac{64dmB^2}{\lambda\epsilon}\right) - 1\right) \cdot \mathbb{E}[F(\mathbf{w}_t) - F(\mathbf{w}^*)] \\
& \quad + 2\eta\left(6\eta + \frac{18}{\sqrt{\lambda m}}\log\left(\frac{64dmB^2}{\lambda\epsilon}\right)\right) \cdot \mathbb{E}[F(\tilde{\mathbf{w}}_s) - F(\mathbf{w}^*)].
\end{aligned}
$$

Recalling that this is an upper bound on $\mathbb{E}[\|\mathbf{w}_{t+1} - \mathbf{w}^*\|^2]$ and changing sides, we get

$$
\begin{aligned}
& 2\eta\left(1 - 6\eta - \frac{18}{\sqrt{\lambda m}}\log\left(\frac{64dmB^2}{\lambda\epsilon}\right)\right) \cdot \mathbb{E}[F(\mathbf{w}_t) - F(\mathbf{w}^*)] \\
& \leq \mathbb{E}[\|\mathbf{w}_t - \mathbf{w}^*\|^2] - \mathbb{E}[\|\mathbf{w}_{t+1} - \mathbf{w}^*\|^2] + \eta\epsilon \\
& \quad + 12\eta\left(\eta + \frac{3}{\sqrt{\lambda m}}\log\left(\frac{64dmB^2}{\lambda\epsilon}\right)\right) \cdot \mathbb{E}[F(\tilde{\mathbf{w}}_s) - F(\mathbf{w}^*)].
\end{aligned}
$$

Summing over all $t = (s-1)T + 1, \ldots, sT$ in the epoch (recalling that the first one corresponds to $\tilde{\mathbf{w}}_s$) and dividing by $\eta T$, we get

$$
\begin{aligned}
& 2\left(1 - 6\eta - \frac{18}{\sqrt{\lambda m}}\log\left(\frac{64dmB^2}{\lambda\epsilon}\right)\right) \cdot \frac{1}{T}\sum_{t=(s-1)T+1}^{sT} \mathbb{E}[F(\mathbf{w}_t) - F(\mathbf{w}^*)] \\
& \leq \frac{1}{\eta T} \cdot \mathbb{E}[\|\tilde{\mathbf{w}}_s - \mathbf{w}^*\|^2] + \epsilon \\
& \quad + 12\left(\eta + \frac{3}{\sqrt{\lambda m}}\log\left(\frac{64dmB^2}{\lambda\epsilon}\right)\right) \cdot \mathbb{E}[F(\tilde{\mathbf{w}}_s) - F(\mathbf{w}^*)].
\end{aligned}
$$

Since $F(\cdot)$ is $\lambda$-strongly convex, we have $\|\tilde{\mathbf{w}}_s - \mathbf{w}^*\|^2 \leq \frac{2}{\lambda}(F(\tilde{\mathbf{w}}_s) - F(\mathbf{w}^*))$. Plugging this in and simplifying a bit leads to

$$
\begin{aligned}
& \mathbb{E}\left[\frac{1}{T}\sum_{t=(s-1)T+1}^{sT} F(\mathbf{w}_t) - F(\mathbf{w}^*)\right] \\
& \leq \frac{\left(\frac{2}{\eta\lambda T} + 12\left(\eta + \frac{3}{\sqrt{\lambda m}}\log\left(\frac{64dmB^2}{\lambda\epsilon}\right)\right)\right) \cdot \mathbb{E}[F(\tilde{\mathbf{w}}_s) - F(\mathbf{w}^*)] + \epsilon}{2\left(1 - 6\eta - \frac{18}{\sqrt{\lambda m}}\log\left(\frac{64dmB^2}{\lambda\epsilon}\right)\right)}.
\end{aligned}
$$

The left hand side equals or upper bounds $\mathbb{E}[F(\tilde{\mathbf{w}}_{s+1})] - F(\mathbf{w}^*)$ (recall that we choose $\tilde{\mathbf{w}}_{s+1}$ uniformly at random from all iterates produced in the epoch, or we take the average, in which case $\mathbb{E}[F(\tilde{\mathbf{w}}_{s+1}) - F(\mathbf{w}^*)]$ is at most the left hand side by Jensen's inequality). As to the right hand side, if we assume

$$\eta = \frac{1}{c} \ , \ \ T \geq \frac{9}{\eta\lambda} \ , \ \ m \geq c\frac{\log^2(64dmB^2/\lambda\epsilon)}{\lambda} \tag{27}$$

for a sufficiently large numerical constant $c$, we get that it is at most $\frac{1}{4} \cdot \mathbb{E}[F(\tilde{\mathbf{w}}_s) - F(\mathbf{w}^*)] + \frac{2}{3}\epsilon$. Therefore, we showed that

$$\mathbb{E}\left[F(\tilde{\mathbf{w}}_{s+1}) - F(\mathbf{w}^*)\right] \ \leq \ \frac{1}{4} \cdot \mathbb{E}[F(\tilde{\mathbf{w}}_s) - F(\mathbf{w}^*)] + \frac{2}{3}\epsilon.$$

Unwinding this recursion, we get that after $s$ epochs,

$$\mathbb{E}\left[F(\tilde{\mathbf{w}}_{s+1}) - F(\mathbf{w}^*)\right] \ \leq \ 4^{-s} \cdot \mathbb{E}[F(\tilde{\mathbf{w}}_1) - F(\mathbf{w}^*)] + \frac{2}{3}\epsilon \sum_{i=0}^{s-1} 4^{-i}$$

$$= \ 4^{-s} \cdot \mathbb{E}[F(\tilde{\mathbf{w}}_1) - F(\mathbf{w}^*)] + \frac{2}{3}\epsilon \cdot \frac{1 - 4^{-s}}{1 - 4^{-1}}$$

$$< \ 4^{-s} \cdot \mathbb{E}[F(\tilde{\mathbf{w}}_1) - F(\mathbf{w}^*)] + \frac{8}{9}\epsilon.$$

Since we assume that we start at the origin ($\tilde{\mathbf{w}}_1 = \mathbf{0}$), we have $F(\tilde{\mathbf{w}}_1 - F(\mathbf{w}^*)) \leq F(\mathbf{0}) = \frac{1}{m}\sum_{i=1}^{m} y_i^2 \leq 1$, so we get

$$\mathbb{E}\left[F(\tilde{\mathbf{w}}_{s+1}) - F(\mathbf{w}^*)\right] \leq 4^{-s} + \frac{8}{9}\epsilon.$$

This is at most $\epsilon$ assuming $s \geq \log_4(9/\epsilon)$, so it is sufficient to have $\lceil \log_4(9/\epsilon) \rceil$ epochs to ensure suboptimality at most $\epsilon$ in expectation.

Finally, note that we had $\lceil \log_4(9/\epsilon) \rceil$ epochs, in each of which we performed $T$ stochastic iterations. Therefore, the overall number of samples used is at most $\lceil \log_4(9/\epsilon) \rceil T$. Thus, to ensure the application of Propositions 1 and 2 is valid, we need to ensure this is at most $m/2$, or that

$$m \ \geq \ 2\lceil \log_4(9/\epsilon) \rceil \cdot T.$$

Combining this with Eq. (27), it is sufficient to require

$$\eta = \frac{1}{c} \ , \ \ T \geq \frac{9}{\eta\lambda} \ , \ \ m \geq c\log^2\left(\frac{64dmB^2}{\lambda\epsilon}\right)T$$

for any sufficiently large $c$. $\qquad\square$

## B  Additional Technical Lemmas

In this appendix, we collect a couple of purely technical lemmas used in certain parts of the paper.

**Lemma 6.** *If $T, m$ are positive integers, $T \leq m$, then*

$$\frac{1}{mT} \sum_{t=2}^{T} (t-1)\left(\sqrt{\frac{1}{t-1}} + \sqrt{\frac{1}{m-t+1}}\right) \ \leq \ \frac{2}{\sqrt{m}}.$$

*Proof.*

$$\frac{1}{mT} \sum_{t=2}^{T} (t-1)\left(\sqrt{\frac{1}{t-1}} + \sqrt{\frac{1}{m-t+1}}\right)$$

$$= \frac{1}{mT} \sum_{t=1}^{T-1} t\left(\sqrt{\frac{1}{t}} + \sqrt{\frac{1}{m-t}}\right)$$

$$= \frac{1}{mT} \left(\sum_{t=1}^{T-1} \sqrt{t} + \sum_{t=1}^{T-1} \frac{t}{\sqrt{m-t}}\right).$$

Since $\sqrt{t}$ and $t/\sqrt{m-t}$ are both increasing in $t$, we can upper bound the sums by integrals as follows:

$$\leq \frac{1}{mT}\left(\int_{t=0}^{T}\sqrt{t}\,dt + \int_{t=0}^{T}\frac{t}{\sqrt{m-t}}\,dt\right)$$

$$= \frac{1}{mT}\left(\frac{2}{3}\cdot T^{3/2} + \left(-\frac{2}{3}\sqrt{m-t}\cdot(2m+t)\right)\Big|_{t=0}^{T}\right)$$

$$= \frac{1}{mT}\left(\frac{2}{3}\cdot T^{3/2} + \frac{2}{3}\left(2m\sqrt{m}-\sqrt{m-T}\cdot(2m+T)\right)\right)$$

$$= \frac{2}{3}\left(\frac{\sqrt{T}}{m} + 2\left(\frac{\sqrt{m}}{T}-\sqrt{m-T}\cdot\left(\frac{1}{T}+\frac{1}{2m}\right)\right)\right)$$

$$= \frac{2}{3}\left(\frac{\sqrt{T}}{m} + \frac{2}{T}\left(\sqrt{m}-\sqrt{m-T}\cdot\left(1+\frac{T}{2m}\right)\right)\right)$$

$$\leq \frac{2}{3}\left(\frac{\sqrt{T}}{m} + \frac{2}{T}\left(\sqrt{m}-\sqrt{m-T}\right)\right)$$

$$= \frac{2}{3}\left(\frac{\sqrt{T}}{m} + \frac{2}{T}\left(\frac{T}{\sqrt{m}+\sqrt{m-T}}\right)\right)$$

$$\leq \frac{2}{3}\left(\frac{\sqrt{T}}{m} + \frac{2}{\sqrt{m}}\right).$$

Since $T \leq m$, the above is at most $\frac{2}{3}\left(\frac{1}{\sqrt{m}}+\frac{2}{\sqrt{m}}\right) = \frac{2}{\sqrt{m}}$ as required. □

**Lemma 7.** *Let $X$ be a random variable, which satisfies for any $\delta \in (0,1)$*

$$\Pr\left(X > a + b\log(1/\delta)\right) \leq \delta,$$

*where $a, b > 0$. Then*

$$\mathbb{E}[X] \leq a + b.$$

*Furthermore, if $X$ is non-negative, then*

$$\sqrt{\mathbb{E}[X^2]} \leq \sqrt{2}\cdot(a+b).$$

*Proof.* The condition in the lemma implies that for any $z \geq a$,

$$\Pr(X > z) \leq \exp\left(-\frac{z-a}{b}\right).$$

Therefore,

$$\mathbb{E}[X] \leq \mathbb{E}[\max\{0,X\}] = \int_{z=0}^{\infty}\Pr(\max\{0,X\}\geq z)\,dz \leq a + \int_{z=a}^{\infty}\Pr(\max\{0,X\}\geq z)\,dz$$

$$= a + \int_{z=a}^{\infty}\Pr(X\geq z)\,dz \leq a + \int_{z=a}^{\infty}\exp\left(-\frac{z-a}{b}\right)dz = a + \int_{z=0}^{\infty}\exp\left(-\frac{z}{b}\right)$$

$$= a + b.$$

Similarly, if $X$ is non-negative,

$$\mathbb{E}[X^2] = \int_{z=0}^{\infty}\Pr(X^2 \geq z)\,dz \leq a^2 + \int_{z=a^2}^{\infty}\Pr(X^2\geq z)\,dz = a^2 + \int_{z=a^2}^{\infty}\Pr(X\geq\sqrt{z})\,dz$$

$$\leq a^2 + \int_{z=a^2}^{\infty}\exp\left(-\frac{\sqrt{z}-a}{b}\right)dz = a^2 + 2b(b+a) \leq 2(a+b)^2,$$

and the result follows by taking the square root. □

# C   Uniform Upper Bound on $F(\mathbf{w}_t) - F(\mathbf{w}^*)$ for SVRG

Below, we provide a crude bound on the parameter $B$ in Thm. 4, which upper bounds $F(\mathbf{w}_t) - F(\mathbf{w}^*)$ with probability 1. Note that $B$ only appears inside log factors in the theorem, so it is enough that $\log(B)$ is reasonably small.

**Lemma 8.** *Suppose we run SVRG (algorithm 1) with some parameter $T$ and step size $\eta \in (0, 1)$ for $S$ epochs, where each $f_i(\cdot)$ is a regularized squared loss (as in Eq. (2), with $\|\mathbf{x}_i\|, |y_i| \leq 1$ for all $i$), and $F(\cdot)$ is $\lambda$-strongly convex with $\lambda \in (0, 1)$. Then for every iterate $\mathbf{w}_t$ produced by the algorithm, it holds with probability 1 that*

$$\log(F(\mathbf{w}_t) - F(\mathbf{w}^*)) \leq 2S \cdot \log(5T) + \log\left(\frac{4}{\lambda}\right).$$

Noting that Thm. 4 requires only $S = \mathcal{O}(\log(1/\epsilon))$ epochs with $T = \Theta(1/\lambda)$ stochastic iterations per epoch, we get that

$$\log(F(\mathbf{w}_t) - F(\mathbf{w}^*)) = \mathcal{O}\left(\log\left(\frac{1}{\epsilon}\right)\log(T) + \log\left(\frac{1}{\lambda}\right)\right)$$

with probability 1.

*Proof.* Based on the SVRG update step, we have

$$\|\mathbf{w}_{t+1}\| \leq \left\|\mathbf{w}_t - \eta\nabla f_{\sigma(t)}(\mathbf{w}_t)\right\| + \eta\left\|\nabla f_{\sigma(t)}(\tilde{\mathbf{w}}_s)\right\| + \eta\left\|\nabla F(\tilde{\mathbf{w}}_s)\right\|. \tag{28}$$

Since we are considering the regularized squared loss, with $\|\mathbf{x}_i\| \leq 1, |y_i| \leq 1$ and $0 \leq \hat{\lambda} \leq \lambda \leq 1$, the first term on the right hand side is

$$\left\|\mathbf{w}_t - \eta\left(\mathbf{x}_{\sigma(t)}\mathbf{x}_{\sigma(t)}^\top\mathbf{w}_t - y_{\sigma(t)}\mathbf{x}_{\sigma(t)} + \hat{\lambda}\mathbf{w}_t\right)\right\| \leq \left\|\left((1 - \eta\hat{\lambda})I - \eta \cdot \mathbf{x}_{\sigma(t)}\mathbf{x}_{\sigma(t)}^\top\right)\mathbf{w}_t\right\| + \eta\left\|y_{\sigma(t)}\mathbf{x}_{\sigma(t)}\right\|$$

$$\leq \left\|\left((1 - \eta\hat{\lambda})I - \eta \cdot \mathbf{x}_{\sigma(t)}\mathbf{x}_{\sigma(t)}^\top\right)\right\|\|\mathbf{w}_t\| + \eta \leq \|\mathbf{w}_t\| + 1,$$

As to the second two terms on the right hand side of Eq. (28), we have for any $i$ by similar calculations that

$$\|\nabla f_i(\mathbf{w})\| = \left\|\left(\mathbf{x}_i\mathbf{x}_i^\top + \hat{\lambda}I\right)\mathbf{w} - y_i\mathbf{x}_i\right\| \leq \left(1 + \hat{\lambda}\right)\|\mathbf{w}\| + 1 \leq 2\|\mathbf{w}\| + 1$$

as well as

$$\|\nabla F(\mathbf{w})\| \leq 2\|\mathbf{w}\| + 1.$$

Substituting these back into Eq. (28) and loosely upper bounding, we get

$$\|\mathbf{w}_{t+1}\| \leq \|\mathbf{w}_t\| + 4\left(\|\tilde{\mathbf{w}}_s\| + 1\right).$$

Recalling that each epoch is composed of $T$ such iterations, starting from $\tilde{\mathbf{w}}_s$, and where $\tilde{\mathbf{w}}_{s+1}$ is the average or a random draw from these $T$ iterations, we get that

$$\|\tilde{\mathbf{w}}_{s+1}\| \leq \|\tilde{\mathbf{w}}_s\| + 4T\left(\|\tilde{\mathbf{w}}_s\| + 1\right) \leq 5T\left(\|\tilde{\mathbf{w}}_s\| + 1\right),$$

and moreover, the right hand side upper-bounds the norm of any iterate $\mathbf{w}_t$ during that epoch. Unrolling this inequality, and noting that $\|\tilde{\mathbf{w}}_1\| = 0$, we get

$$\|\tilde{\mathbf{w}}_{S+1}\| \leq (5T)^S \cdot 1 = (5T)^S,$$

and $(5T)^S$ upper bounds the norm of any iterate $\mathbf{w}_t$ during the algorithm's run.

Turning to consider $\mathbf{w}^* = \arg\min_{\mathbf{w}} F(\mathbf{w})$, we must have $\|\mathbf{w}^*\|^2 \leq 1/\lambda$ (to see why, note that any $\mathbf{w}$ with squared norm larger than $1/\lambda$, $F(\mathbf{w}) \geq F(\mathbf{w}^*) + \frac{\lambda}{2}\|\mathbf{w}\|^2 > \frac{1}{2}$, yet $F(\mathbf{0}) = \frac{1}{2m}\sum_{i=1}^m y_i^2 \leq \frac{1}{2}$, so $\mathbf{w}$ cannot be an optimal solution). Moreover, $F(\cdot)$ is 2-smooth, so for any iterate $\mathbf{w}_t$,

$$F(\mathbf{w}_t) - F(\mathbf{w}^*) \leq \|\mathbf{w}_t - \mathbf{w}^*\|^2 \leq (\|\mathbf{w}_t\| + \|\mathbf{w}^*\|)^2 = \left((5T)^S + \sqrt{\frac{1}{\lambda}}\right)^2.$$

Since $(5T)^S$ and $\sqrt{1/\lambda}$ are both at least 1, this can be upper bounded by $\left(\frac{2(5T)^S}{\sqrt{\lambda}}\right)^2 = \frac{4}{\lambda} \cdot (5T)^{2S}$. Taking a logarithm, the result follows. $\quad\square$

# D Pseudocode of Distributed Without-Replacement SVRG

---

**Algorithm 2** Distributed Without-Replacement SVRG

---

   **Parameters:** $\eta, T$

   **Assume:** $\{f_1(\cdot), \ldots, f_m(\cdot)\}$ randomly split to machines $1, 2, \ldots, n$ (possibly different number at different machines)

   Each machine $j$ splits its data arbitrarily to $b_j$ batches $B_1^j, \ldots, B_{b_i}^j$ of size $T$

   $j := 1$ , $k := 1$ , $t := 1$

   All machines initialize $\tilde{\mathbf{w}}_1$ at $\mathbf{0}$

   **for** $s = 1, 2, \ldots,$ **do**

      Perform communication round to compute $\tilde{\mathbf{n}} := \nabla F(\tilde{\mathbf{w}}_s) = \frac{1}{m} \sum_{i=1}^m \nabla f_i(\tilde{\mathbf{w}}_s)$

      Machine $j$ performs $\mathbf{w}_1 := \tilde{\mathbf{w}}_s$

      **for** Each $f$ in $B_k^j$ **do**

         Machine $j$ performs $\mathbf{w}_{t+1} := \mathbf{w}_t - \eta \left( \nabla f \mathbf{w}_t \right) - \nabla f(\tilde{\mathbf{w}}_s) + \tilde{\mathbf{n}} )$

         $t := t + 1$

      **end for**

      Machine $j$ computes $\tilde{\mathbf{w}}_{s+1}$ as average of $\mathbf{w}_1, \ldots, \mathbf{w}_T$, or one of them drawn uniformly at random.

      Perform communication round to distribute $\tilde{\mathbf{w}}_{s+1}$ to all machines

      $k := k + 1$

      If $k > b_j$, let $k := 1, j := j + 1$

   **end for**

---