[Reviews · NeurIPS 2016]

Reviewer 1

Summary

The paper studies the problem of minimizing the average of a finite sum of convex functions over a convex domain using stochastic algorithms that, opposed to most popular methods, apply WITHOUT-replacement sampling to the data. More specifically, the paper considers methods that first randomly permute the functions, and then process the functions via incremental updates one at a time, making at most a single pass over the data (hence the data is only shuffled once). There are three main motivations for considering stochastic methods with without-replacement sampling: 1. it is observed many times empirically (and to the best of my knowledge this is what ML practitioners really do) that applying random shuffles to the data and then apply incremental updates works better than with-replacement SGD. 2. After the data is randomly permuted, these algorithms require only sequential access to the memory, which is much more efficient than standard with-replacement SGD methods that require random access to perform the updates. Since the main setting under consideration here is when making at most a single pass, it sounds plausible to assume that the data is already stored in some random permutation, and hence the algorithm is fully sequential, and there is no need to even artificially permute the data. 3. In a distributed setting in which data is partitioned to several machine (not assuming the data is sampled i.i.d from a distribution), it is more natural and efficient to analyze without-replacement algorithms. While this setting is well motivated there has not been a large body of work with strong positive results on it which is understandable, since analyzing without-replacement sampling algorithms is much more difficult than the with-resampling variants. Once such previous recent result focused on the regime in which the algorithm makes a large number of passes over the data, re-permuting the data before each new pass. This current paper focuses on a regime which I find more interesting and relevant to ML applications, in which at most a single pass is made over the data. Indeed the celebrated with-replacement SGD method is most interesting exactly in this regime (though suffering from the short-coming of with-replacement sampling discussed above). The main results are as follows: 1. the paper shows that any algorithm for which one can give a regret bound in the online learning adversarial model (which holds for SGD for instance) can be applied in the setting above with convergence rate resembling the average regret bounds. In particular, when all functions are convex Lipschitz losses for a linear predictor, wo-replacement SGD exhibits roughly the same convergence rate as standard SGD. 2. when the average over all functions is also strongly convex, wo-replacement SGD gives an improved rate, again similar to standard SGD up to log factors. 3. The authors gives a linearly converging wo-replacment variant for the celebrated SVRG algorithm in the specific yet important case in which the loss functions are the squared-loss. 4. The authors use the previous result to improve the application of SVRG for distributed optimization in the wo-replacement setting, improving the regime of parameters (i.e. the condition number) for which the method is highly efficient (in terms of communication).

Qualitative Assessment

As I detailed above, I believe the setting under consideration (without-replacment sampling for finite sum optimization using single pass) is very well motivated and widely applicable in ML problems. The results, to the best of my knowledge, are very novel and will be potentially interesting to many in the community, both theorists and practitioners. The paper is also very well written and results are carefully explained. Some minor comments: -when defining F_{1:t-1}, I think it will also be beneficial for the reader to add a subscript of \sigma, reminding the reader this function is random. - in Lemma 1, please define s_{1:t-1} and s_{t:m}. - in Theorem 2, in the centered equation, it should be \mathcal{R}_{t-1,m-t+1}, i.e., comma instead of colon. - first eq. in proof of Theorem 3: on the RHS it should be the squared norm of w_{t+1}.

Confidence in this Review

3-Expert (read the paper in detail, know the area, quite certain of my opinion)


Reviewer 2

Summary

The authors give a convergence analysis of stochastic gradient algorithms for convex functions on convex domains using a without-replacement-scheme. They show that in certain cases (convex Lipschitz loss of linear prediction, \lambda strongly convex), the guarantees of with-replacement sampling can be matched. In the last part of the paper, without-replacement sampling is also applied to SVRG and the authors show that under certain conditions only one shuffling is necessary. Furthermore, they give an application to distributed learning and argue that without replacement sampling can be beneficial in this case.

Qualitative Assessment

Though it seems very simple, the comparison of with-replacement sampling and without replacement sampling has proven a difficult problem, which connects to certain open problems in linear algebra. There has been some recent substantial progress, and the paper is following up on this, covering different parameter regimes. This is an important area of research and the paper is a significant contribution. The presentation of the results is very clear, the results themselves are remarkable and the proofs in the appendix are formulated in a way easy to comprehend, despite their comlexity. Some minor typos should be corrected: - l. 128: f needs to be convex, otherwise the subgradient is not well-defined. - l. 134: transductive - Theorems 1 and 4: Is w^* as in l.131? - l. 180: Delete "be" - l. 224/226: Square should be outside

Confidence in this Review

2-Confident (read it all; understood it all reasonably well)


Reviewer 3

Summary

This paper provides convergence guarantees for stochastic gradient methods, under several cases where the number of passes over the data is small, and also when no data reshuffling is necessary. These guarantees under various conditions (Theorems 1, 2, 3 4) are new and establish that pertinent SG methods with limited replacement sampling are not significantly worse than those with-replacement sampling.

Qualitative Assessment

I like your paper. I did not find any mistakes. However I did not go over section A in detail.

Confidence in this Review

2-Confident (read it all; understood it all reasonably well)


Reviewer 4

Summary

The paper explores the theoretical properties of sampling without replacement in stochastic gradient methods. The author shows that the convergence rate is the same as sampling with replacement provided the number of stochastic gradient steps is less than the sample size. The analysis covers the algorithms with o(T) regrets, including stochastic gradient descent (SGD), and stochastic variance reduced gradient (SVRG) methods for regularized least square problem. For the former, the convergence rate is T^-0.5 generally and log T / T for strongly convex functions. For the latter, the convergence is guaranteed under technical assumptions. The author novelly uses Transductive Rademacher complexity as the tool and provide elegant proofs for the results.

Qualitative Assessment

It is a great pleasure to read this paper. From my perspective, sampling without replacement is really important in practice but is not well studied in optimization literatures. The idea that uses transducive Rademacher complexity is very smart and novel. It provides a fundamental tool to analyze sampling without replacement. In addition, the matrix Chernoff bound for sampling without replacement used to deal with least square problem is also appealing and reveals the possibility to solve more general problems. I believe that the new technique will encourage more people to explore the sampling without replacement. The paper is very well-written and clear. Although technical, the presentation makes the results easy to understand. In addition, the appendix is my favourite part since it is logical and detailed. There are some minor flaws: 1) in line 224 and 225, L + muB^2 should be (L + mu B)^2; 2) the explanation between line 230 to 240 intends to explain the technical parts but not easy to follow without reading the proofs, maybe expand more details or simply delete it.

Confidence in this Review

2-Confident (read it all; understood it all reasonably well)


Reviewer 5

Summary

This paper discussed about the stochastic gradient methods using without-replacement sampling in contrast of prevailing literature about SGD using with replacement sampling. The methods utilized in this paper is different from other existing papers about SGD without-replacement sampling. The result of performance guarantee is weaker than [8], but it brings another interesting perspective of understanding SDG without-replacement. It also proposed a distributed SVRG with using SDG without-replacement in inner iteration, which is more reasonable choice than with-replacement (at least to me).

Qualitative Assessment

# following words are written in latex-format In general, this paper is well-written, gives a novel proofs and presents several interesting results. I have several following concerns. \begin{enumerate} \item Theorem 1 looks very weird to me.\\ 1) In the right hand side of equation after line 159, the summation $t$ is ranging from 2 to $T$. However, this doesn't make sense in later term $F_{t:m}(w_t)$, when $t$ is larger than m. I think it should be $T$ mod by $m$. Also in introduction, you mention the performance of a single pass. But in section 3, you said {\em any} sequence of T convex Lipschitz. Here {\em any} means the order or any $T$ and any order?\\ 2) The theorem is based upon the assumption that there is a regret bound $R_T$. If I understand correctly, we can directly conclude that when $T$ is the multiple of $m$, we have \[ \Ex\left[\frac{1}{T}\sum_{t=1}^T F(w_t) - F(w^\star)\right] = \Ex\left[\frac{1}{T}\sum_{t=1}^T [f_{\sigma(t)}(w_t) - f_{\sigma(t)}(w^\star)]\right] \leq \frac{R_T}{T} \] Why bother to do this extra lossier bound?\\ 3) Even if this theorem aimed to give the bound on the any $T$, this conclusion still cannot conclude SGD without-replacement sampling won't be worse too much than SGD with-replacement sampling since unknown $R_T$(Especially, when some new algorithm is proposed, how could you apply it? And [8] already give a nice boundary on SGD-type algorithm.). What is the meaning of theorem 1/corollary 1 exactly?\\ 4) According to appendix, the theorem 1 is not the necessary foundation for theorem 3 (I think theorem 3 is much more reasonable and interesting). Therefore, I have the feeling that theorem is not only unnecessary but also not meaningful.\\ Maybe I have some misunderstanding. Please clarify theorem 1. \item In algorithm 1 description, should you restrict $T$ to be the multiple of $m$ or any $T$ is feasible? \item Theorem 3. Would it be better to emphasize "if we run without-replacement sampling SGD" than "if we run SGD"? \item Algorithm 4. Several typos: line 4 should be $B^j_{b_j}$ instead of $B^j_{b_i}$. The notation of inner loop doesn't make sense to me. Please fix it (such as $\nabla fw_t)$). Is each machine run sequentially(incremental) or paralleled? Please give more explanation and detail about that. I think this is more useful than theorem 1. \end{enumerate}

Confidence in this Review

2-Confident (read it all; understood it all reasonably well)


Reviewer 6

Summary

In stochastic gradient implementations, usually the data examples are shuffled once before each pass (epoch) begins. This is effectively without replacement sampling, and in practice it usually works better than with replacement sampling. However, existing theoretical results on SGD are based on with replacement sampling, and without replacement sampling is rarely discussed because of lacking of analysis tools. This paper tries to show that without replacement gradient methods, during one pass over data, has the same suboptimality bound (in terms of T, the number of samples visited) as with-replacement gradient methods in general cases (Convex Lipschitz and strongly convex functions). The differences between this work and previous papers are: (1) this work applies to general convex functions (not just least square); (2) it does not require the with replacement gradient methods to run at least k passes to get better convergence guarantee than without replacement methods; the guarantee in the paper applies to the first pass over the data; (3) this work does not try to show without replacement methods are better with a large number of passes, rather, it shows without replacement methods performs similar (with a similar suboptimality) as with replacement methods even during a single pass. The paper also extends its analysis to a recently very popular stochastic gradient method - SVRG. The author is able to show that under certain conditions, for the least square problem, SVRG can converge linearly with only a single shuffle at the beginning. This results can help distributed SVRG implementations work better by using less data exchanges between worker machines.

Qualitative Assessment

This paper is overall well written, and the theoretical results are solid. The proofs for Lipschitz convex case in this paper decompose the expected bound to two parts: the regret, and the other part can be bounded by Lemma 1, and then bounded by Rademacher complexity. Lemma 1 in this paper is an important result, and is the key for all following proofs. This framework is great for analyzing other problems involving without replacement operations. This paper does not contains experiments, but I think it's OK, because people know without replacement gradient methods work better in practice, and previous papers contain those results. But for the SVRG case, because this result is relatively surprising, some experiment results will make this paper even stronger. One weakness is that the without replacement SVRG analysis is only for Least Square, limiting its usefulness. However I believe the results in this paper will inspire more researchers look into this problem.

Confidence in this Review

2-Confident (read it all; understood it all reasonably well)